Manuscript prepared for Atmos. Chem. Phys.
with version 2014/07/09 7.01 Copernicus papers of the LaTeX class copernicus.cls.
Date: 14 July 2016

# Estimation of the advection effects induced by surface heterogeneities in the surface energy budget

**J. Cuxart**[1], **B. Wrenger**[2], **D. Martínez-Villagrasa**[1], **J. Reuder**[3,4], **M.O. Jonassen**[3,5], **M.A. Jiménez**[1], **M. Lothon**[6], **F. Lohou**[6], **O. Hartogensis**[7], **J. Dünnermann**[2], **L. Conangla**[8], and **A. Garai**[9]

[1]Universitat de les Illes Balears, Palma (Mallorca, Spain)
[2]Hochschule Ostwesfalen-Lippe, Höxter (Nordrhein-Westfalen, Germany)
[3]Geophysical Institute, University of Bergen (Norway)
[4]Bjerknes Centre for Climate Research, Bergen (Norway)
[5]The University Centre in Svalbard (Norway)
[6]Centre Recherches Atmospheriques, Univ. Toulouse & CNRS, Lannemezan (France)
[7]Wageningen University of Research (The Netherlands)
[8]Universitat Politècnica de Catalunya, Manresa (Catalonia, Spain)
[9]University of California, San Diego (California, USA)

*Correspondence to:* J.Cuxart (joan.cuxart@uib.cat)

**Abstract.** The effect of terrain heterogeneities in one-point measurements is a continuous subject of discussion. Here we focus on the order of magnitude of the advection term in the equation of the evolution of temperature as generated by documented terrain heterogeneities and we estimate its importance as a term in the surface energy budget (SEB), for which the turbulent fluxes are computed using the eddy-correlation method. The heterogeneities are estimated from satellite and model fields for scales near 1 kilometre or broader, while the smaller scales are estimated through direct measurements with remotely-piloted aircraft, thermal cameras and also by high-resolution modeling. The variability of the surface temperature fields is not found to decrease clearly with increasing resolution, and consequently the advection term becomes more important as the scales become finer. The advection term provides non-significant values to the SEB at scales larger than few kilometres. On the contrary, surface heterogeneities at the metre scale yield large values of the advection, which are probably only significant in the first centimetres above the ground. The motions that seem to contribute significantly to the advection term in the SEB equation in our case are roughly those around the hectometre scales.

## 1 Introduction

The Surface Energy Budget (SEB) is the expression of the conservation of energy for a volume across the atmosphere-surface interface, which should take into account all the energy exchanges taking place in it. Traditionally (see e.g. Oke, 1987, or Foken, 2008a, 2008b) it is expressed as an equilibrium equation between the Net Radiation ($Rn$) -usually the larger term- and the three other principal terms, the turbulent sensible heat flux ($H$), the latent heat flux ($LE$) and the soil heat flux ($G$). Conceptually, as described in Moene and Van Dam (2014) or Cuxart et al. (2015), it is computed for a layer of infinitesimal depth across the interface in a horizontally homogeneous area, therefore no storage or source terms are considered and, formally, the budget is expressed as

$$Rn + H + LE + G = 0 \qquad (1)$$

where a possible criterion for the signs is that they are positive if they are directed towards the surface. This approach, when brought to the experimental field, implies a number of practical difficulties, which we may reduce to two major issues. Firstly, the impossibility of measuring a differential volume at the interface. As Foken (2008a) exemplifies, each instrument is measuring a signal corresponding to a different volume of air. One way to overcome this conceptual difficulty is to acknowledge that we measure in a volume limited by the position of the instruments used. This implies that we must account for storages and look for the divergence of the heat fluxes across the volume of measurement. Besides, the heat sources and sinks of energy within the volume must also be included (such as the energetic effects of biologic and anthropic activities).

The second issue comes from the fact that the Earth's surface is not homogeneous. To check the validity of equation (1), one should look for flat homogeneous locations, therefore distant from topographical features (even minor ones) or from changes in the soil uses (like different crops close to each other). These terrain heterogeneities may induce turbulent eddies and change the values of the turbulent heat flux compared to a completely homogeneous area.

The need of the scientific community to make experimental measurements, even in complex terrain, implies that these limitations should be progressively overcome. Another important factor to consider is that instrumental errors in the determination of the turbulent fluxes must be kept in mind, very often implying an underestimation of their value, due to the non-capturing of certain scales by the measuring devices (Foken 2008a). All taken into account, lead Foken (2008b) to acknowledge that, to progress in our understanding of the physics of the surface-atmosphere exchange, we must resign ourselves to work with imbalances of the order of 20% in Equation (1).

Cuxart et al. (2015) derive a complete SEB equation from the evolution equation of the temperature of a volume. They take a conceptual box with the top at the screen level and the bottom just under the surface. Simplifying the equation accordingly, they produce a budget equation for the volume where the turbulent fluxes are located at screen level, the conduction flux just under the surface, the advection terms can be computed using the divergence of temperature across the volume limits and the missing terms can be accounted for explicitly if the information is available (see Figure 1 in that paper). The rationale in that paper leads to an extended SEB equation:

$$Rn + H + LE + G + S + B + TT + A + Ot = 0 \tag{2}$$

where each term is considered at its own position relatively to the interface. Here $S$ stands for the effect of the sources and sinks in the volume, including the storage in the mass elements and $B$ the energy exchanges linked to the biologic and anthropic process (Moene and Van Dam, 2014 and references therein), $TT$ is the tendency of the temperature and $A$ describes the effect of the advection term, presumably linked to the heterogeneities of the surface . The term $Ot$ represents any other effect not accounted for in the budget, including the instrumental errors.

This approach is still insufficient because it implies several oversimplifications, such as not considering the internal variability of the volume, like the presence of objects over the ground or soil heterogeneity, or some inputs from outside the volume, like water pumped up from below the volume of interest by plant roots (Moene and Van Dam, 2014). All these effects are gathered into $Ot$, which is not estimated. Nevertheless, it accounts explicitly for some elements of the imbalance, trying to progress with some insight in the SEB approach used in many practical environmental applications.

The total imbalance is expressed by Cuxart et al. (2015, eq. 7) as the sum of the contributions of the tendency, the storage, the biological processes, the advection effects and the other unaccounted factors ($Imb = S + B + TT + A + Ot$). The imbalance values, using long term averages, is usually between 10% of the net radiation in flat homogeneous conditions (Oncley et al., 2007), increasing with the terrain complexity to more than 30%. For individual averages of some minutes, the values can become much larger.

In this work we concentrate on the importance of the advection term $A$ in the SEB. which represents the effect of the motions of timescales longer than the turbulence-averaged ones. Short-lasting surface temperature homogeneities induce eddy motions that are, in essence, turbulent, and therefore just treated statistically. If the inhomogeneities last significantly longer that the averaging time to compute the turbulent fluxes then, by construction from the Reynolds decomposition, their effect has to be taken into account by the advection term. The latter may be expressed as (in W m$^{-2}$)

$$A = \rho Cp \Delta z \sum_{i=1}^{3} u_i \frac{\Delta T}{\Delta x_i}. \tag{3}$$

The related timescale must be the one used for the computation of the other terms in the budget. If 30-minute averages are used for the radiation or the turbulent heat fluxes, then the increments of temperature must be computed using 30-minute averages of temperature. Coherent structures lasting longer than this averaging time are most likely contributing significantly

to this term, as would be the case for circulations between adjacent parcels of terrain at different temperatures, of a spatial scale still to be determined.

The term has some arbitrariness, especially in the value that must be taken for the dimensions of the box $\Delta x_i$. We will consider here thermal surface heterogeneities that last significantly longer than the characteristic times of the turbulence so that they can be treated as persistent thermal surface patterns generating durable circulations near the surface. It is unclear which are the scales contributing to the term without being distorted by obstacles between the measurement site and the different heterogeneities under consideration, as it is discussed in Leuning et al. (2012).

The BLLAST experiment (Lothon et al., 2014) provided the opportunity of gathering several teams with different experimental and modeling expertises at the Lannemezan Plateau (Gascony, France) in summer 2011. In this work, we analyze data from different sources operating during BLLAST with the aim of estimating the order of magnitude of $A$. At this point, it is necessary to make clear that reliable quantitative conclusions are very difficult to obtain with the approach used in this work and the available data. However, comprehensive qualitative results will be obtained based on broad approximations and estimations of the order of magnitude of $A$ depending on the scale analyzed. Therefore, we consider it a first methodological step opening the way to more precise and focused studies to come.

As mentioned above, a number of simplifications are made to collectively treat a large amount of heterogeneous information. We will confine our estimations of $A$ to providing an order of magnitude of the term, taking $1\ \mathrm{m\ s^{-1}}$ as the characteristic wind speed at 2m above ground level (a.g.l.) for the analyzed events, which is a good approximation for the observed values (not shown). The standard deviation of the surface temperature will be used as a surrogate of the average horizontal temperature gradient, as supported by the measurements of the remotely controlled multicopter during the BLLAST campaign.

The concept of a "footprint" is not used in this work because the area is composed of patches of different land-use with a characteristic size of 100 m in all directions and this approach would be difficult to implement considering 30-minute averages (for a discussion see Foken and Leclerc (2004) or, more recently, Hartogensis, 2015). The average vertical wind speed is taken as zero, acknowledging that this implies neglecting vertical advection, therefore implicitly included in the $Ot$ term. Finally the concept of a "blending height" is only used sporadically since, as Foken (2008a) indicates, it may be not very appropriate when analyzing the effects of heterogeneity at relatively small scales.

In Section 2 the different sources of information are described, highlighting their potentialities and limitations. This is followed in Section 3 by a description of the SEB for the period June 30 to July 3 and an analysis of the method. Section 4 provides a short description of the estimates of $A$ for scales of the order of a kilometre to those of the order of a metre. In Section 5 an overall discussion of the findings is given before presenting the conclusions in Section 6.

## 2 Tools

During BLLAST a large number of teams contributed instrumentation. While the main purpose of the experiment was to study the late afternoon and evening transition regimes of the atmospheric boundary layer(Lothon et al., 2014), a second objective was to understand the effect of small-scale terrain heterogeneities in the boundary layer. This paper focusses on the latter goal. Ideally one should compare a perfectly homogeneous location with an inhomogeneous one, the former being actually very difficult to find over land, at least in mid-latitudes. The approach taken here, as described in the Introduction, is to use the available data to estimate the value of the advection term corresponding to the existing heterogeneities as detected by various observations.

BLLAST had two supersites, at site 1 there were vertical profiling devices, including radiosondes, and a number of surface layer measurements, some intended to assess the effect of the surface heterogeneities. Site 2 was intended to study well defined heterogeneities measuring over corn, moor and forest sites, each of an approximate scale of 1 km, larger than the average heterogeneities on the Lannemezan Plateau (van de Boer et al., 2014). The small-scale experiment under analysis here took place in site 1.

A complete SEB station was installed by the universities of Bergen and the Balearic Islands over a square of 160 m side over which there was previously a radar, currently installed at a nearby location. This "small square" is at a first look approximately flat and homogeneous, but a closer inspection shows that there is a very smooth slope towards SW and that the vegetation cover is irregular, some small areas being covered by grass and dead grass, others are bare and most of them are a mixture of short grass and bare soil.

The small square was surrounded by areas of grass for cattle, some wooden spots and fields of different crops, essentially the same landscape surrounding the area for several kilometres, with the exception of the city area of Lannemezan. The average scale of each of these landscape units was a few hundreds of metres at most, and most typically they had 100 m of characteristic size.

The BLLAST campaign was characterized by the passage of weather fronts approximately each third or fourth day, with clear skies and weak pressure gradients in between, when the wind dynamics over the Lannemezan plateau were dominated by the upvalley and downvalley circulations from the nearby Vallée d'Aure (Jiménez and Cuxart, 2014). Therefore, the moisture availability at the surface was high, resulting in large daytime evapotranspiration fluxes (Bowen ratios below 1). We will focus in this work on the anticyclonic period between the rainy events of June 29 and July 3, which includes three BLLAST Intensive Observational Periods.

## 2.1 Measurements in the small square

**Surface Energy Budget station:** In the small square, the SEB consisted of i) the full radiation balance (Kipp and Zonen CNR1) at 0.8 a.g.l., ii) the sensible and latent heat fluxes with a Campbell Scientific integrated system that includes a CSAT-3 sonic anemometer and a Licor-7500 fast open-path $CO_2$ and water vapor concentration sensor, at 1.95 m a.g.l., sampling at 20 Hz, iii) three Hukseflux ground soil flux plates at a depth of 5 cm, sampling at 1 Hz, and, iv) intermittently, a vertical array of eight thermocouples sampling at 10 Hz displayed at heights logarithmically spaced between 0.015 and 1.92 m a.g.l.

**Upper soil moisture content:** From June 21 to July 2 manual measurements of the upper 5 cm of soil moisture were made at selected spots (30, separated roughly 30 m on distinct areas, such as grass, half bare or short natural vegetation) within the small square once per day during Intensive Observing Periods. The sensors were Delta-T ML3 devices with a 1% accuracy. At each spot, several measurements (typically 5) were taken in an area of 2m of radius due to the high variability of the measurements and the final data for each spot was the average value. A gentle slope towards SW favors accumulation of water at this part of the small square after rainy events.

**Multicopter:** From 1 to 5 July, a team of the Hochschule Ostwestfalen-Lippe operated a remotely-piloted multicopter (Wrenger et al., 2013; Jiménez et al., 2016) over the small square, making 27 transects following a defined pattern that covered most of the well-defined land-use areas, at a height near 7 m a.g.l. In a limited number of cases, profiles were made up to 50 m. Here, we use only the air temperature data -as sampled by a fast thermocouple supplemented with a slower Sensitron SHT75 sensor- and the surface temperature, provided by a factory-calibrated sensor Melexis MLX90614. The IR sensor had a view angle of 35° implying a resolution of about 5 m, if the sensor is vertically looking, and degrading to about 10 m when the flight was slightly tilted in respect to the horizontal.

**UCSD multispectral camera:** A multispectral camera of the University of California at San Diego (Garai and Kleissl, 2013) was mounted at a height of 50 m a.g.l. in the 60-m tower close to the NW of the square. It was pointing to the square from June 30 and was able to produce fields of surface temperature of the NW portion of the square with a time resolution of 1s, a spatial resolution of 0.38 m $\times$ 0.18m with field of view 92m $\times$ 52m. The rated accuracy is 0.08 K.

**WUR multispectral camera:** On June 21, a team of the Wageningen University measured surface temperature with a thermal camera within the small square producing high-resolution (centimetre scale) fields. The sensor was a IR Snapshop camera from Infrared Solutions, producing images with 120 x 120 pixels and a field of view of 17 degrees, taken from a height of 1.5 m a.g.l.. Each field was for a rectangle of approximately 3 x 2 $m^2$. The procedure took a couple of hours, it started at mid-afternoon with clear skies and ended an hour before sunset, with overcast skies that brought rain shortly after.

## 2.2 Measurements, model and satellite estimations at hectometre and larger scales

**SUMO:** The Small Unmanned Meteorological Observer (Reuder et al., 2009; Reuder and Jonassen, 2012) is a small (0.8 m of wingspan) remotely controlled aircraft that was operated by the University of Bergen. For the purposes addressed here, it was flown at an approximate height of 65 m a.g.l., over a squared area of roughly 1.6 km on a side. The small square is about one-hundredth of the total area of the "SUMO square". The flights followed a grid pattern and, although each flight covered a slightly different area, the small square area was always included. Here, only temperature data are used. The air temperature was measured by a Sensirion SHT 25 sensor, mounted inside a radiation protection tube on the wing, while the surface temperature was estimated with a MLX90614 IR sensor, which had an angle of view of approximately 90°, with an effective resolution at the ground close to 100 m. The typical ground speed was around 20 m s$^{-1}$. The SUMO operations during the BLLAST campaign and the data processing of the surface temperature data are described in detail in Reuder et al. (2016).

**Meso-NH:** The simulation outputs of this non-hydrostatic model (Lafore et al., 1997) are used. The run was from June 29 at 0000 UTC to July 3 at 0000 UTC, considering the first six hours as the spin up period, using three domains (Figure 1), with the same physical options and vertical discretization as the simulation for the same area used in Jiménez and Cuxart (2014). The external domain (D1) covers the SW part of France, including the Pyrenees and the Western valleys of the Massif Central, at a resolution of 2 km. The second domain (D2) is over the Central Pyrenees and the plain at the foothills with a resolution of 400 m, while the inner domain (D3) has a resolution of 80 m for a square of 250 grid points each side, and is centered over the

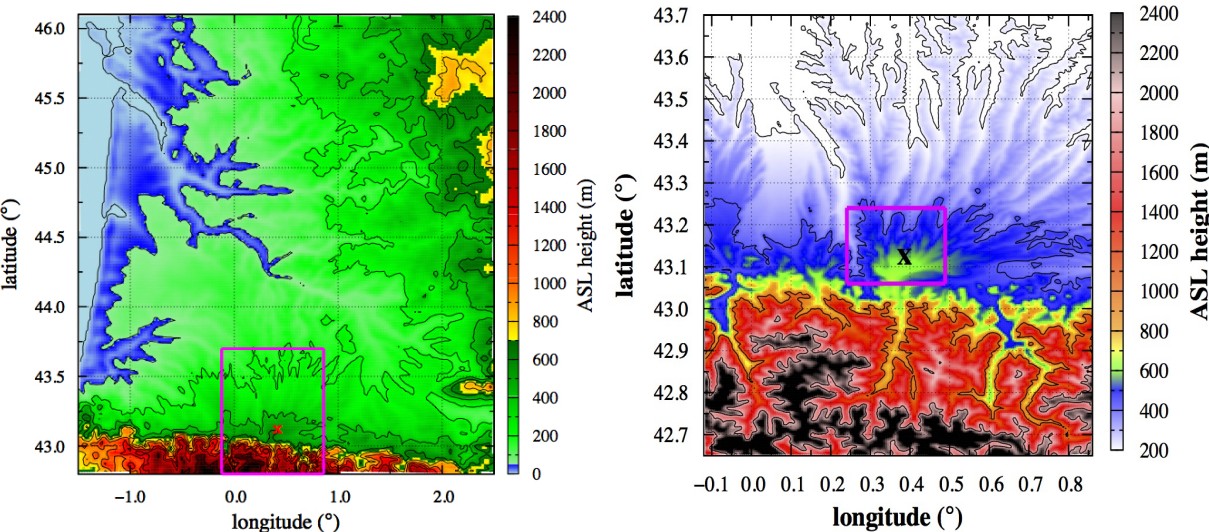

**Figure 1.** Left: Topography of southwestern France which corresponds to the larger domain of the model simulation (D1). The inner area inside the purple square corresponds to domain D2. Right: Domains D2 and D3. The cross indicates the location of Lannemezan. Surface temperatures for areas with heights above sea level between 50 and 700m (in green in Left Figure) are used to compute the average LST and its standard deviation.

small square covering approximately the Lannemezan Plateau. D2 was only run between 1800 UTC of July 2 to 1000 UTC of July 3, and D3 only between 0000 UTC and 1000 UTC of July 2, due to limited availability of computational resources.

The model uses a standard one-dimensional turbulence 1.5 order scheme in the three domains (Cuxart et al., 2000), the ISBA soil scheme (Noilhan and Planton, 1989) and the radiation scheme of Morcrette (1990) as the more relevant parameterizations. It is initialized with the analysis of the ECMWF for June 29 at 0000 UTC and is run until 0000 UTC of July 3, with lateral boundary conditions provided as well by the ECMWF. A sponge layer is activated at its top. The numerical estimations of the surface temperature field and of the air temperature at different heights are used.

**Satellite data:** The cloud-free areas of the satellite images from the MODIS sensor (Salomonson et al., 1989) onboard the Terra and Aqua polar orbit satellites, available between June 30 and July 3, 2011 for SW France, are used to compute the standard deviation of Land Surface Temperatures (LST) as estimated by these radiometers with a resolution at this latitude of about 1 km. In very stable conditions, LST may present some uncertainties (Martínez et al., 2010), mostly related to condensation and frost on the surface elements, that may change their emissivity values. Meteosat Second-Generation (MSG, Schmetz et al., 2002) data at a resolution near 5 km is also used to provide time series of average LST and its standard deviation with a time resolution of 15 minutes. Unlike the MODIS images, MSG LST is not corrected for atmospheric water vapor.

### 2.3 Treatment of the advection term

As a first guess, it will be assumed that the depth of the volume for which the SEB will be computed is 2 m, since this is the typical distance between measurements at the surface layer and at the ground, that allow computation of vertical divergences. The horizontal dimension of the box will be the subject of this work, since we will explore what would be the contribution of the advection term to the budget depending on the horizontal scale of the thermal heterogeneity. It is clear that these computations will be rough estimates of the effect of the advection in the SEB, but it provides a reasonable starting point.

For simplification purposes, we will

- neglect the vertical advection (taking $w = 0$ in average is reasonable), implying that the associated error is included in the $Ot$ term of the complete SEB;

- take $1 \mathrm{\,m\,s^{-1}}$ as the order of magnitude of the wind in the Surface Layer for clear skies and non windy cases, the subject of this study, regardless of its direction, therefore ignoring the sign of $A$;

- approximate the average horizontal surface temperature gradient in an area by the standard deviation of the surface temperature, supported by SUMO measurements, keeping in mind that we are concerned solely with orders of magnitude of $A$;

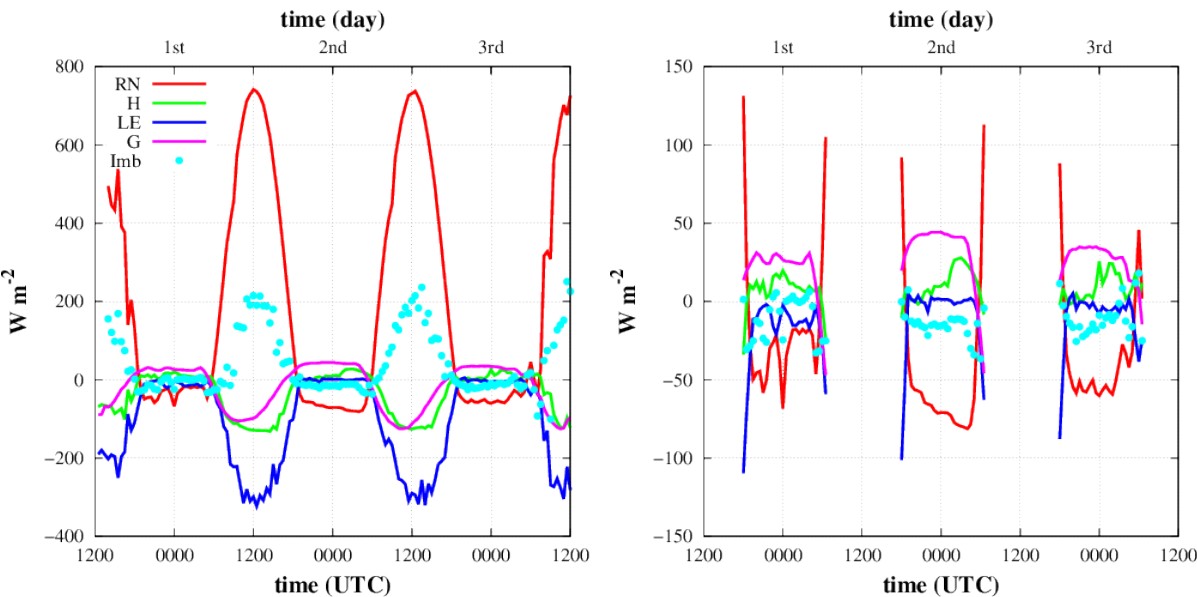

**Figure 2.** Surface energy budget in the small square for the period June 30 to July 3, between two rainy events (left), and zoom for the nighttime periods (right).

– consider the LST variability as a good estimation of the variability of the air temperature at the Surface Layer, as supported by the measurements of the multicopter;

– take the factor $\rho C p \Delta z\, u \approx 2500\ J(K\,m\,s)^{-1}$, where $\Delta z = 2$ m, leading to an expression for the order of magnitude of the advection term.

It is clear, from the large number of hypotheses made and its significance, that the results presented below will be broader estimations of the value of $A$ for a given scale and source of information, with large uncertainties of the order of 100% or even above. However, these results will show significant differences in the orders of magnitude for the explored scales, leading to a number of informative results. The approximate equation that we will use reads

$$O[A] \approx 2500 \frac{\Delta T}{\Delta x}. \tag{4}$$

## 3   The measured Surface Energy Balance in the small square

The SEB is computed for the station in the small square for the period 30 June at 1200 UTC to 3 July at 1200 UTC, between two rainy events. It shows a progressive drying of the upper soil as it will be shown in the next section. The evolution of the different terms of the budget for the whole period is shown in Figure 2.

The turbulent fluxes are computed every 30 minutes using the eddy-correlation method with the standard corrections, the same used for all equipments in BLLAST (De Coster and Pietersen, 2011), using the EC-pack (van Dijk et al., 2004) that includes the computation of the planar fit angles to virtually rotate the sonic into the mean flow (Wilczak et al., 2001) and the Webb correction for the fluctuations of density (Webb et al., 1980). Errors in the values of the turbulent fluxes are estimated to be approximately 10%. Furthermore, the ensemble of the BLLAST data set was quality controlled. This includes verifying record timing and de-spiking.

The net radiation term in eq. (1) is the result of the budget of the four terms measured by the CNR1 (longwave and shortwave upward and downward fluxes). While the ground flux is measured at 5 cm below the surface, corrections are required due to unrealistic measured values of the upper soil temperature. Specifically, corrections are made to surface temperature values using harmonic analysis (Heusinkveld et al., 2004) and simplifying the heat flux to a single sinusoidal function (Hillel, 1998). This correction results in an average increase in the ground heat flux of 40% and a delay of 90 minutes.

Similar to what is done in Cuxart et al. (2015), we consider positive terms those giving energy to the volume and negative those extracting energy from it. In this 72-hour series, shown in Figure 2, we see that in the daytime, $Rn$ is the only input of

energy and this energy is transported vertically away from the surface by turbulent latent and sensible fluxes and downward through the ground flux. However, there is an excess of incoming energy that is not accounted for by these processes. This daytime imbalance is similar in magnitude to the latent heat flux, and larger than the sensible and the ground heat fluxes. In the small scale square, the amount of vegetation is small, so the $B$ term is not important. Moreover, there are no clear sources or sinks, or objects with storing capacity, meaning that $S$ is also expected to be small. The tendency is found to be of the order of a few W m$^{-2}$. Therefore, this imbalance, which is larger than 100 W m$^{-2}$ during the mid-day hours, must be attributed to the advection term $A$ or to other unaccounted processes or factors in $Ot$.

Figure 2 also displays a zoomed-in view of the budget for the three nights. Taking the second night for discussion, which is the one showing the most smooth time evolution, $Rn$ is the largest term, now a loss, and the compensating heat fluxes are $G$ and the $H$, whereas $LE$ is very small. The imbalance is approximately one-fourth of $Rn$, around 20 W m$^{-2}$. In general terms, depending on the wind intensity, latent heat can be released through condensation or captured through evaporation. Again, since $TT$, $B$ and $S$ seem to be irrelevant, the imbalance should be explained either by $A$ or $Ot$.

In the following sections, we will explore, using the available modeling and observational information, the order of magnitude of the values of the advection term in the SEB, making use of the observed horizontal temperature gradients in equation (4).

## 4 Estimation of the advection term at various scales

### 4.1 Scales between 1 and 10 kilometres

The order of magnitude of the advection term at scales close to 1 kilometre or larger can be estimated using model outputs and satellite data. The green color in Figure 1 show the areas in domains D1 and D2 where the terrain has a height above sea level between 50 and 700 m. This selection avoids coastal areas and mountainous terrain, so that the terrain complexity is comparable to that around Lannemezan. The average values of LST and air temperature at some levels and the standard deviations are computed for these areas in green. The same statistics are computed from the available LST provided by MODIS (about 4 per day) and MSG (every 15 min).

Figure 3 shows the evolution of LST as seen by the model, MODIS on Aqua and Terra satellites, and SEVIRI on MSG for domains D1 and D2. Also, the average values of the air temperature at 1.5, 10, 50 and 100 m above the surface are shown. It is noteworthy to point out that the model and satellite LST values are comparable, allowing to use the model statistics with some confidence.

In Figure 3, we see that at a resolution of 2 km (D1) the standard deviation value of the air temperature does not change with height, varying between 1 K and 2 K with maximal values in the afternoon and minimal at the end of the night. The LST standard deviation is higher, with values around 3 K in the day and 2 K in the night. These values are from the three different available sources (D1, MODIS and MSG). Note that large sporadic values of standard deviation for MSG on June 29 are due to cloud passages. Taking the 1.5-m values of the standard deviation as an approximation to the typical changes of temperature over 2 km, the advection term according to equation 4 has a rough order of magnitude of less than 5 W m$^{-2}$ both for night and day, for scales of 2 km or larger.

For the higher resolution run D2, we see that the standard deviation decreases with height, indicating that at this resolution the model is able to react significantly to the prescribed surface variability. The model has the largest values of the standard deviation at the 1.5-m level, varying between 0.7 and 1.8 K for the series shown. The corresponding rough order of magnitude according to equation 4 is less than 10 W m$^{-2}$. While the values of the 1.5-m air temperatures are very close in both model domains, the standard deviation is larger in the domain at lower resolution. This is probably indicating that higher horizontal resolution is able to transport more efficiently heat differences originated at the surface level.

Therefore, for scales larger than 1 km the expected contribution of the advection term to the SEB would be of the order of 10 W m$^{-2}$ in the daytime and 5 W m$^{-2}$ at night, but its relative contribution is much smaller for the daytime than for the night. The sign of the advection term could be determined from inspection of the wind direction between heterogeneities. However, for simplicity, we restrict ourselves to discuss the order of magnitude of the term.

### 4.2 Scales under 1 kilometre

Small scale thermal heterogeneities may generate corresponding small scale circulations. If these patterns are short lived (few minutes), the corresponding circulations can be considered turbulence, but if they are relatively persistent (longer than the averaging time for the computation of the turbulent fluxes) then these circulations should contribute to the advective term in the equation of T or, equivalently, in the SEB equation. Here we will analyze the variability of the temperature fields from the different available sources, and see how the advection term would behave at these fine scales.

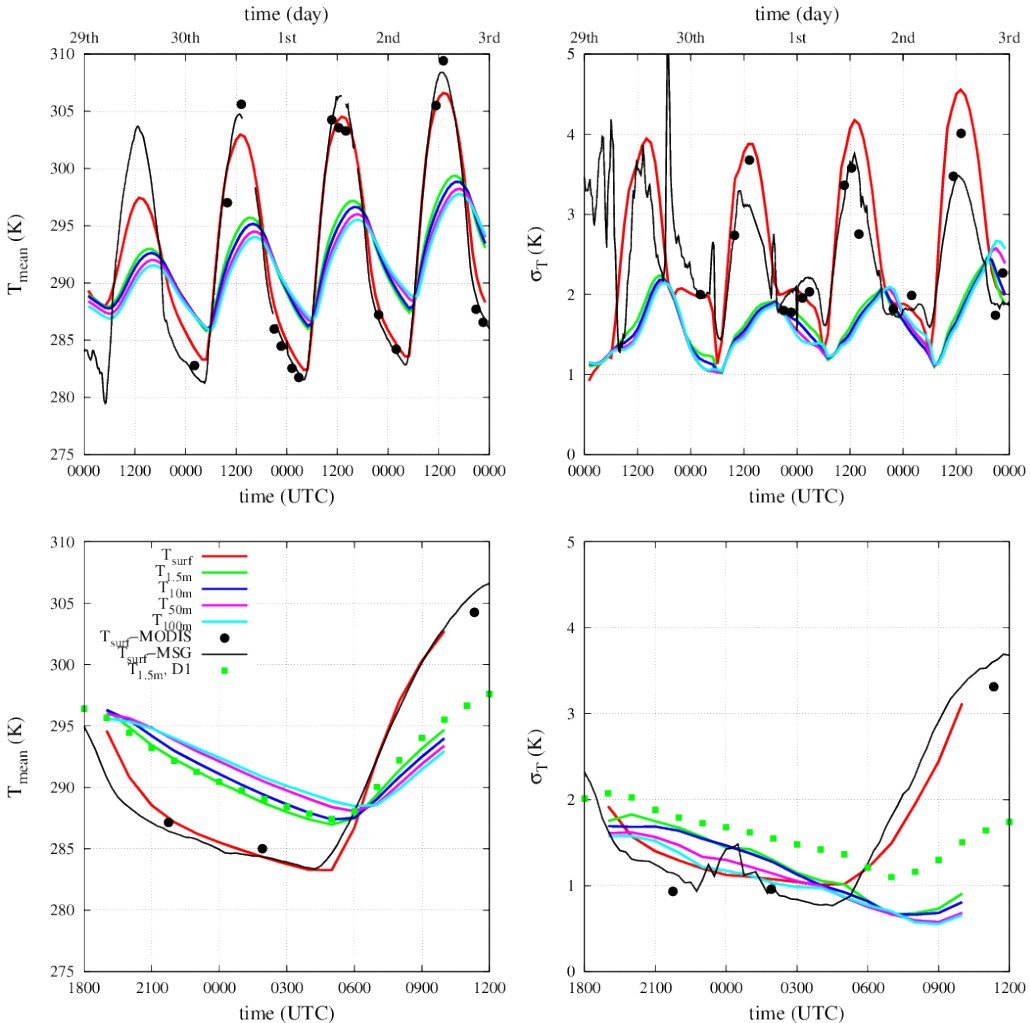

**Figure 3.** Top left: evolution of the average air temperature for some levels of the model and LST, the available MODIS images and Meteosat Second Generation for model domain D1 for a period of 4 days, Top right: standard deviation of the same variables in D1. Bottom left and right: as above for domain D2, which was run only for the night 2 to 3 July. Values for the Temperature at 1.5-m average and standard deviation at D1 are given for comparison in the bottom figures.

#### 4.2.1  The hectometre scale as seen by SUMO

As mentioned previously, SUMO flew at approximately 65 m AGL, over a square of 1.6 km of side (the "SUMO square"), from sunrise to shortly after sunset. It provided, among other data, values of air temperature at that height and of LST sampled at 1 Hz, respectively at resolutions of 10 and 100 m, the latter with overlapping areas, always including the small scale square at Site 1, with a flight duration typically of 10 minutes. Figure 4 shows two typical examples of the LST, one in the afternoon when the small square is warmer than its surroundings and one for the evening, when the small square does not show a significant departure from the average value of the area. With the horizontal resolution of the IR-sensor being close to the size of the small scale heterogeneities site, the related thermal contrasts are probably underestimated.

If we split the measured LST by SUMO in two categories, one from inside the small square and one from outside, we can compute thermal differences, as shown in Figure 5 (left panel), the site warms during the first 5 hours of the day (up to 5 K), and the difference slowly decreases afterwards from 1000 UTC until sunset, when it becomes negative and has values of about -1 K for the next hour. On the other hand, at a height near 65 m, the contrast is very weak or non-existent (Figure 5, right panel), showing that the effect of the thermal differences at the surface almost vanishes somewhere under this level.

If we estimate the order of magnitude of the advective term in the SEB (equation 4 using LST differences), the advection term is less than 10 W m$^{-2}$ just after sunrise and increases to about 60 W m$^{-2}$ at the instant of maximum temperature

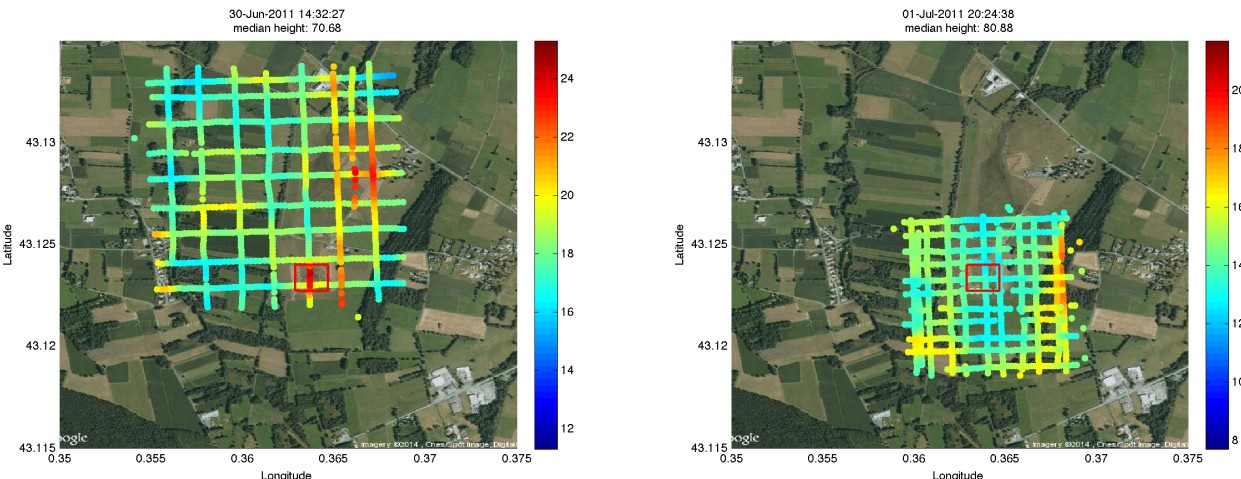

**Figure 4.** Surface temperature on June 30, 2011 at 1432 UTC (left) and on July 1 at 2024 UTC (right) as measured by SUMO from an approximate height of 65 m AGL. The red rectangle indicates the position of the small square of 160 m of side, where many surface layer measurements were made. The units in the colourbar are degrees centigrades.

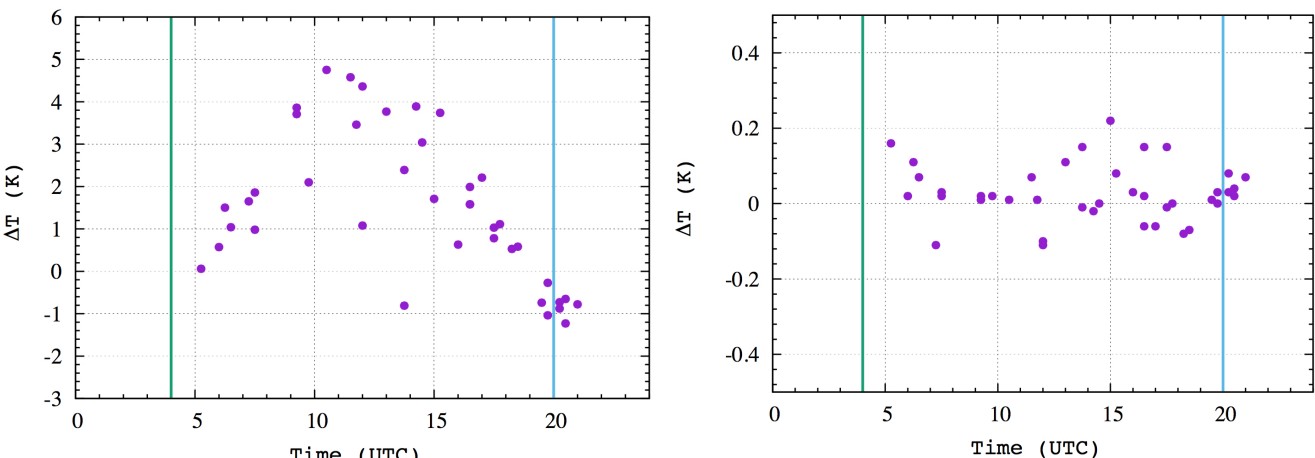

**Figure 5.** Mean air temperature difference between the SUMO square measurements taken inside and outside the small square for all flights during the whole BLLAST campaign, displayed by hour of the day for the LST (left) and the air temperature at approximately 65 m a.g.l. (right). Vertical lines indicate approximate sunrise (0420 UTC) and sunset (1940 UTC).

difference between the square and the surroundings, slowly decreasing to about 10 W m$^{-2}$ near sunset. These values may be overestimations since they have been computed using LST instead of the temperature of air at 2m.

A very important result is that the standard deviation of LST ($\sigma(LST)$, Figure 6, left panel) for the complete SUMO square has a very similar time evolution as the one of the difference of temperatures between the small square and the average of the SUMO square (an estimation of $\Delta(LST)$). The factor of proportionality varies between 1 (in the morning and the evening) and 2 (at the centre of the day). Since we are concerned with orders of magnitude, a factor 2 allows to take $\sigma(LST)$ as a surrogate of $\Delta(LST)$. We shall keep this fact in mind, since we will apply it to some other sources based on this experimental evidence, recalling that the variability of LST is considered as an acceptable surrogate of the air temperature in the Surface Layer, as it will be seen later with the multicopter data.

### 4.2.2 The hectometre scale as seen by the Meso-NH model

The Meso-NH model with domain D1, covering the Garonne basin and surroundings, was run for four days of the BLLAST campaign, whereas D2 and D3, covering respectively the Lannemezan plateau and surroundings and the SUMO square and

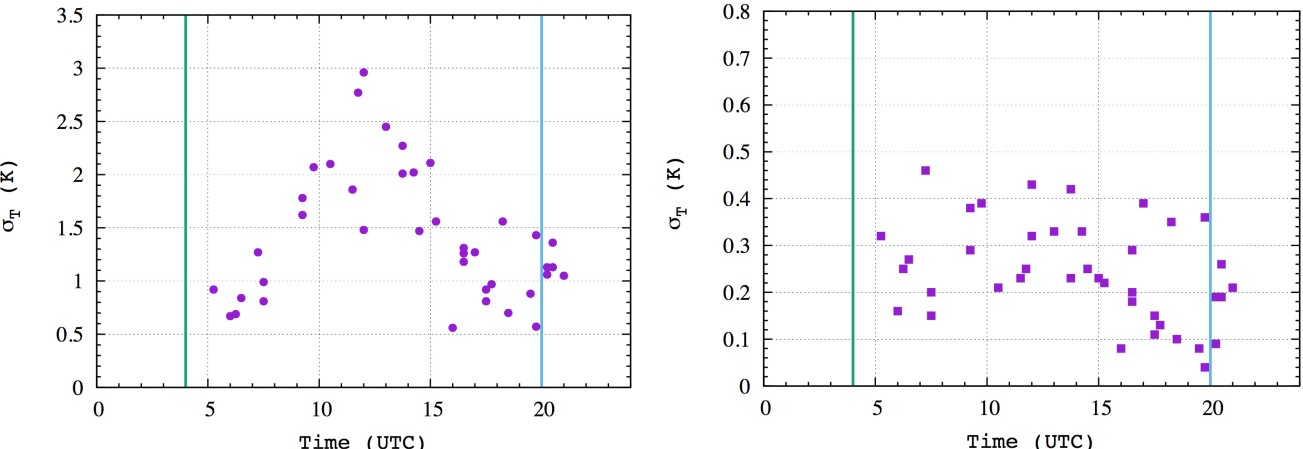

**Figure 6.** Standard deviation for the SUMO square for all flights during the whole BLLAST campaign, displayed by hour of the day for the LST (left) and the air temperature at approximately 65 m AGL (right). Vertical lines indicate sunrise (0420 UTC) and sunset (1940 UTC).

surroundings were only run for the night 1 to 2 July due to the limitations of computational resources (D2 from 1800 UTC to 1000 UTC and D3 from 0000 UTC to 1000 UTC). Using a model at high-resolution has the advantage of having all the model information for the relevant variables, whereas the main disadvantage is the unavoidable departure from observations; in our case, the variability of prescribed surface characteristics may be distant from reality.

The SUMO square is described by 25 model columns in D2 and by 441 model columns in D3 (D1 has insufficient resolution to provide variability for the area). During the available hours for D2, $\sigma(LST)$ is close to 0.6 K from sunset to late night and falls to 0.3 K to shortly after sunrise. Then, in the morning hours it increases linearly to values close to 1.3 K at 10 UTC (Figure7 left). The behavior of D3 is similar at night, but after sunrise it allows for higher values (0.7 K) and the increase is slower having only 1 K at 10 UTC (Figure 7 right).

The variability for air temperature is explored taking four levels (1.5, 10, 50 and 100 m AGL) for every column of the SUMO square in both domains (Figure 7). The standard deviation diminishes with height to values around 0.2 K at 50 m independently of the hour, allowing to consider that the effect of the surface heterogeneities is mixed by convection in the daytime or does not reach these heights in the nighttime.

In the surface layer, represented here by the 1.5 and 10 m model levels, we see that the standard deviations are similar to the
ones for the surface temperature in the nighttime (about 0.6 K) but significantly smaller in the daytime (0.4 K compared to about 1 K for the LST), when turbulence manages to reduce the differences effectively. This indicates that it is a fair approximation to take $\sigma(LST)$ as $\sigma(T)$ during the night and in the morning and evening transitions, but that only half of its value should be taken during the daytime. Hereafter, based on these results, we make the strong assumption that we can approximate the surface layer temperature variability using the LST, which is possible in the frame of this study due to our aim to simply provide estimates
of the order of magnitude of $A$.

Therefore, estimating the advection terms with these standard deviations (taking 0.5 K for the whole day in both domains) we get values of the order of 2 W m$^{-2}$ for the larger domain and of order of 15 W m$^{-2}$ for the smaller domain, due to the decreasing value of $\Delta x$. D3 provides values comparable to those computed from the SUMO at night and smaller during the day. Probably, these values are also underestimated due to smaller variability in the model LST than in reality.

**4.2.3    The decametre resolution estimated using soil moisture measurements**

For several days during BLLAST, instantaneous point measurements of superficial soil moisture (SM, defined as the percent of water in the soil volume) were made inside the small square using manual Delta-T devices, that provided an integrated value for the layer between the surface and -5 cm. For each measuring spot, several measurements were taken in an area with a radius of about 2 m, and the average value was saved. Variability at this fine scale was high (Evett et al., 2006). The square had a very
gentle slope (less than 1%) towards the southwest (SW) corner, where rain water tended to accumulate superficially. The soil texture is mostly clay, and mostly covered by grass (alive and dead), with some bare spots.

Figure 8 shows the progressive and inhomogeneous drying over the square for three days after the rainy event of June 29. The day after the rain (June 30) most of the square has values of SM above 30% increasing westwards to more than 40% and with water over the ground in the SW corner, with SM close to 60%. During the second day (July 1), there is a progressive

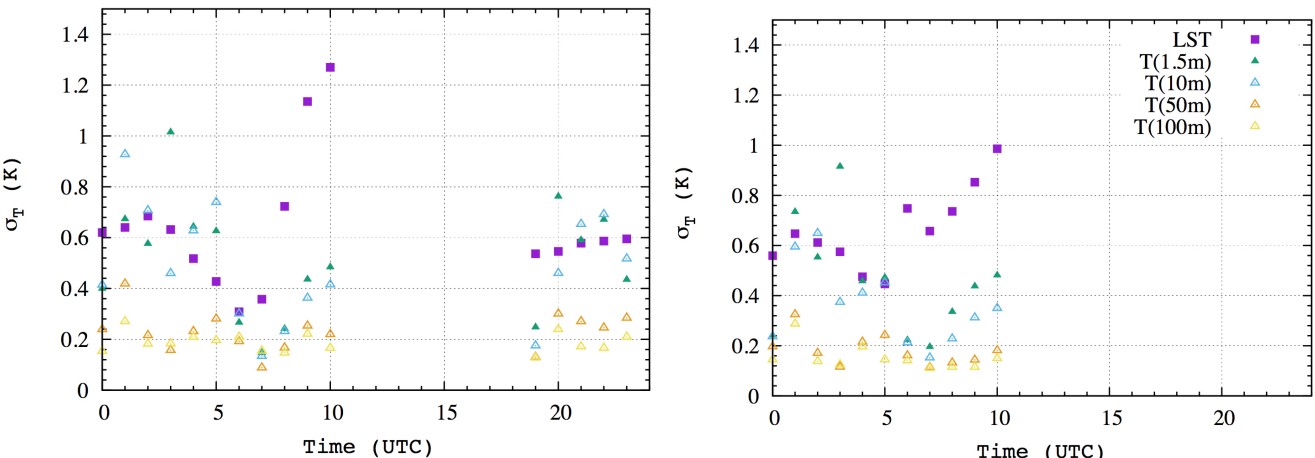

**Figure 7.** Evolution of the standard deviation of temperature over the SUMO square computed from D2 (left, run only between July 1 at 1800 UTC and July 2 at 1000 UTC) and from D3 (right, run only between 0000 and 1000 UTC of July 2 )

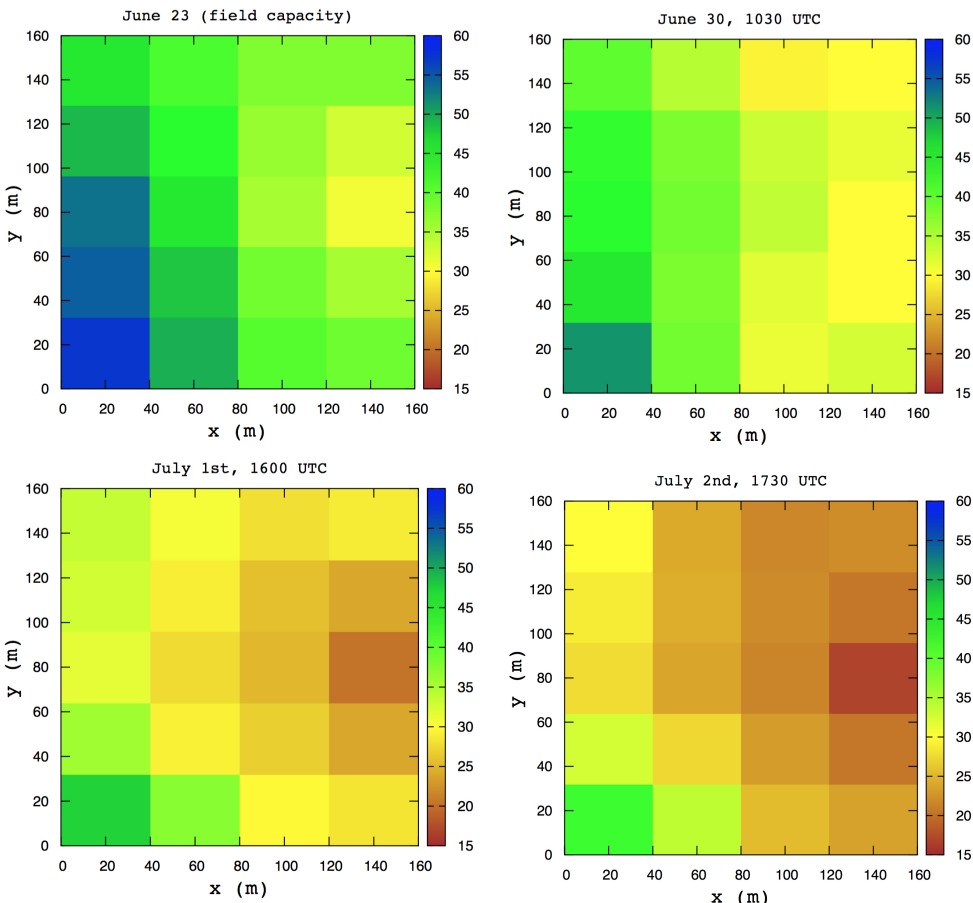

**Figure 8.** Maps of the soil moisture (first 5 cm, in percent of volume) in the small square derived from point measurements from June 30 to July 2. Measurement on June 23 is included for reference, since that day the terrain was at field capacity.. Units of the colorbar are percent of volume.

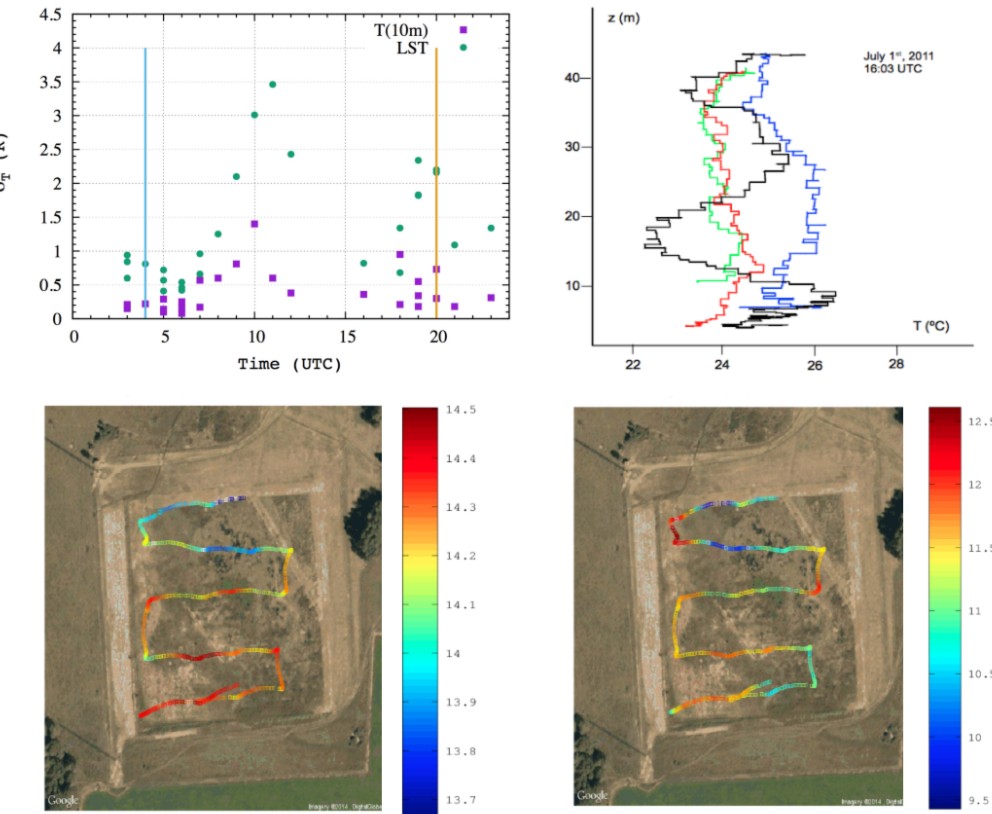

**Figure 9.** Multicopter: LST and air temperature standard deviation for the small square for the ensemble of flights during 5 days, blue and purple lines indicating respectively sunrise and sunset times (top left); vertical profiles, in a different color for each nearby position, inside the small square in the afternoon, made within 5 minutes in a sunny afternoon (top right). Nocturnal flight pattern and air temperature at 5 m a.g.l (bottom left) and LST values from that height (bottom right) for the flight at 0325 UTC of July 5, 2011. Units in the colorbar are degree Celsius.

drying and the bare soil areas on the eastern side (E) have reduced values of the SM of less than 20%, whereas the western (W) part has values between 25 and 60%. These heterogeneities imply very different values locally of the Bowen ratio and of the surface temperature. On the third day (July 2) drying continues, but the surface variability is similar along the square.

This information is not transported into any quantitative estimation - this will be done in the next subsections using IR sensors. However, it may be deduced from these observations that heterogeneities at the decametre scale are large and of longer timescale than the turbulent motions. They may force a relatively steady distribution of eddies inside the square, diminishing the representativity of any point measurement within it.

#### 4.2.4    The decametre resolution as sampled with the multicopter

The OWL multicopter flew over the small square in the period 1 to 5 July at different times of the day. Flights were of short duration (several minutes) and consisted of horizontal transects at an approximate height of 5 m. In addition, some vertical profiles were made up to about 30 m a.g.l. over some selected points. The spatial resolution of the LST measurements from this height, with a cone of view of the IR sensor of 40° is around 5 m, and the fact that the flights have some inclination with respect to the horizontal make 10 m a more conservative spatial resolution estimation. The air temperature is sampled at 1 Hz, equivalent to a spatial resolution of a few metres. The relatively slow response time of the sensor is compensated by a numerical correction scheme which assumes a linear response of the sensor for the difference between instantaneous measured variables (here, air temperature) and the true ambient value of this parameter (Reuder et al., 2009).

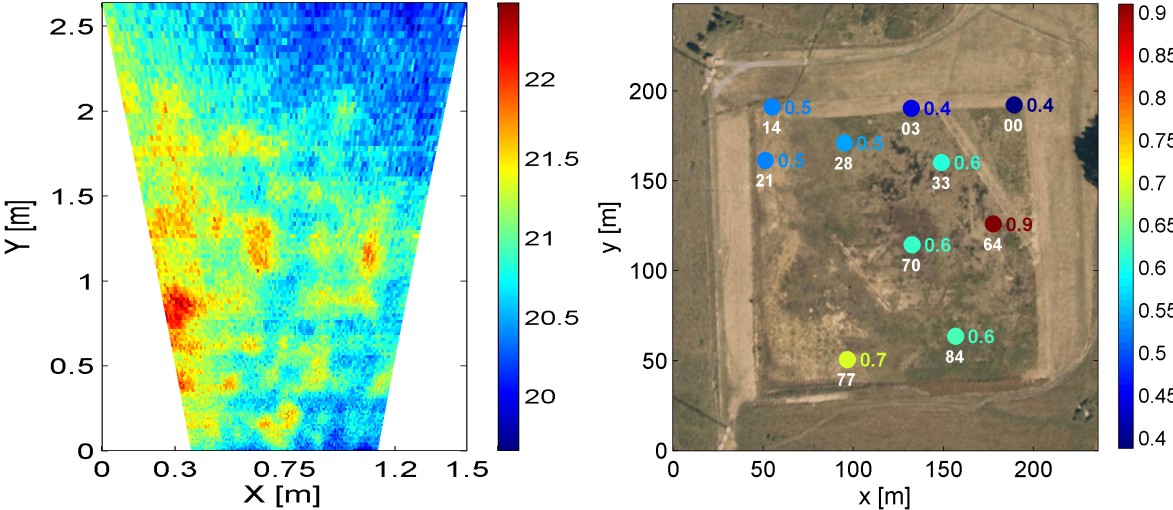

**Figure 10.** Variability inside a WUR IR image for a grass area at 1820 UTC on June 21 (left); standard deviation of LST for 10 similar measurements in the small square from a height of 1.5 m AGL during the evening of the same day, the numbers in white are minutes from the start of measurements, those made after minute 33 were after sunset (right). Units in the colorbar are degree Celsius.

Figure 9 displays the standard deviations of the air temperature and of the LST as a function of the hour of the day, each point corresponding to a flight made during the five days. The values for LST are between 0.5 and 1 K late at night and during the morning transition. They increase to 3.5 K during the morning decreasing to values essentially between 1 and 2 K in the evening transition. The pattern is very similar to what has been found from the model and the SUMO data, but the values are larger except for the late night and morning transition. Some large values of $\sigma(LST)$ just before the sunset are noteworthy.

Here $\sigma(T(5m))$ has the particularity that it is obtained at levels that can be readily compared to model data at 10 m. The model computed values of $\sigma(T)$ were of the order of 0.5 K for D3, and the multicopter measurements are of the same order in the daytime and close to 0.3 K in the nighttime and the transitions. The profiles, shown in Figure 9, indicate that the variability in this afternoon case are of the order of 1-2 K, whereas the few available night profiles (not shown) have differences of some tenths of K, comparable to the respective $\sigma(LST)$. As an example of the method, LST and T(5 m) transects are also displayed in Figure 9 for a late night flight, and we can see that surface and 5 m temperatures show similar patterns of variability, the latter having a smaller amplitude. The qualitative behaviour of the standard deviations of LST and the air temperature in the Surface Layer is very similar, suggesting that variabilities of LST and air temperature in the Surface Layer are comparable when computing orders of magnitude, which is one of the major hypotheses of this work.

If we translate these estimations of $\sigma(T)$ into the advection term, they provide values larger than those estimated for the hectometre scales, because they take place on smaller scales $\Delta x$. Estimating the values from Figure 9 as $\Delta T$ equal to 0.5 K for the day and to 0.2 K for the night, the corresponding advection values would be 100 and 40 W m$^{-2}$. It is not possible to know at this stage if these thermal differences are transient, and therefore their effects taken into account in the turbulence fluxes, or more sustained in time, although the latter case seems not likely.

### 4.2.5 The metre resolution as seen by thermal imagery

**IR sensor at 1.5 m AGL**: Figure 10 (left panel) displays an image taken with the WUR IR Snapshot camera over a grass area in the small square. This surface typically contains green grass, dead grass and some small spots of bare soil, a typical example of the surface of the area. LST is patchy with up to 2 K variations in less than 1 m of distance. In Figure 10 (right panel), it can be seen that the standard deviation of LST is of the order of 0.5 K regardless of the type of surface inspected, whether it is daytime with clear skies (measurements between minutes 00 and 33) or cloudy at nighttime (from minute 64). Probably no organization of the flow can exist at such small scales, but associated microcirculations could exist that would break homogeneity, which can be a key factor in the night, also opposing to the runaway cooling effect as it may be experienced in some numerical models (Viterbo et al., 1999).

**IR sensor at 50 m AGL**: The USDC IR sensor pointing at the NW corner of the small square provides LST fields at a spatial resolution of 0.4 m x 0.2 m approximately. As indicated in the analysis of the overall results in Garai and Kleissl (2013), the average value of the standard deviation for this area is around 0.3 K for 30-min averaging period. They report that, in

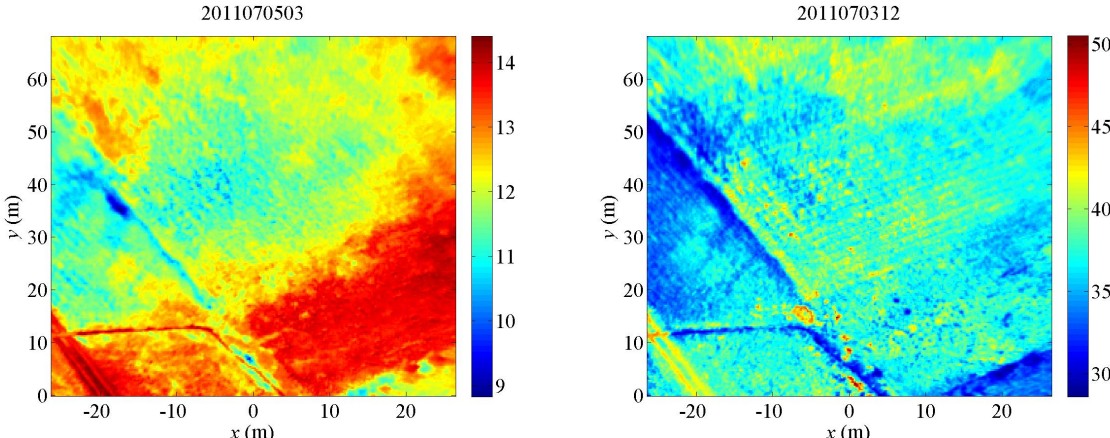

**Figure 11.** NW corner of the small square as seen from the USCD IR camera mounted at 50 m at 0300 UTC on July 5 (left) and 1200 UTC (right) on July 3, 2011. The picture is oriented in a way that the top part looks to SE and the axes are different from those of Figure 10. Units in the colorbar are degree Celsius. .

the daytime, the high and low temperature structures seem to be highly correlated with ejection and sweep events. Figure 11 displays one image during the night of July 5 and one at noon of July 3. The night image shows that in the small square (tones yellow to blue) there are variations of 1-2 K at several scales, whereas the warm area -in red outside the square- may induce for this night a larger-scale circulation. During the day, the amplitude of the differences is larger (of the order of 5 K) and the patches seem to be of smaller scale. The moisture contents at the upper part of the soil may modulate these variations, but in general there was good availability of water in the upper part of the soil due to recent rain events. Albedo may also change significantly with the changes of SM, decreasing as SM increases (Sugathan et al., 2014).

## 5   Discussion

To help making a compact discussion, a summary of the previous results is given in Table I. Let us recall that the main aim of this work, provided the available methods and data, is to provide comprehensive qualitative results for $A$ for each analized scale, hoping that more precise experiments will be conducted in the near future. To proceed, we estimate the gradient of temperature $\Delta T/\Delta x$ as $\sigma(T)/r$, where $r$ stands for the resolution. As described, two strong hypotheses are behind this approximation. Firstly, we can use $\sigma(T)$ as a surrogate for $\Delta T$ which is based on the comparison of Figures 5 (left panel) and 6 (left panel), which show that the time evolution and amplitude of $\sigma(T)$ for the SUMO square compare well with $\Delta T$ between the small square and the rest of the SUMO square.

Secondly, as seen in Figure 3 (bottom right) for the 400 m resolution run of Meso-NH and, more clearly, in Figure 9 (top left), when comparing the standard deviations of the air temperature in the surface layer and of LST obtained by the multicopter on the small square, it is reasonable to assume that the standard deviation of these quantities (temperature of air in the surface layer and LST) have values of the same order of magnitude. These approximations exclude that we can provide meaningful values for the advection term $A$ of the SEB, but it allows for the estimation of generic orders of magnitude, which is a first step towards the increase of understanding. A final reminder is that the mean vertical velocity at the temporal scale of our SEB (30-minute averages) is taken as zero, therefore any vertical advection is neglected and will be accounted implicitly in the term $Ot$ of equation (2).

An important issue to mention is that the uncertainties inherent to each method should be considered in Table 1, even if they are already conceptually taken into account in the term $Ot$ of equation 2. The model, as seen in Figure 3, has an error for our case not larger than 1 K, as it is also the case for most remote sensing determinations of the surface temperature (see, e.g., Coll et al. (1995) for MODIS). Thermal cameras report uncertainties of the order of 0.1 K.

One obvious result is that the order of magnitude of the advection term increases as the scale becomes finer. Therefore the usual assumption that this term is very small compared to the main ones of the SEB equation stands for scales as fine as one kilometre or broader. This is in agreement with the previous argumentations of Foken (2008a, 2008b) and Leuning et al. (2012), that indicated that the advection term was not large enough to explain a substantial part of the imbalance in the measured SEB using only the 4 main terms (equation 1).

**Table 1.** Estimation of the order of magnitude of the advection term $A$ in the Surface Energy Budget equation, for different sources and scales, taking 200 W m$^{-2}$ as imbalance at the center of the day (D) and 30 W m$^{-2}$ at night (N), also considering the error of each source. The orders of magnitude are rounded, as are the percents of the imbalance. Standard deviation of LST values are used as surrogates of horizontal gradients of the Surface-Layer air temperature.

| Source | Scale r (m) | D/N | $\sigma(T)(K)$ | $O(\sigma(T)/r)(K/m)$ | $O(Adv(T))(W\,m^{-2})$ | % Imb |
|---|---|---|---|---|---|---|
| Model D1 and MSG | 4000 | D | 3 | 0.00075 | 2 | 1 |
|  |  | N | 2 | 0.0005 | 1 | 4 |
| Model D2 and MODIS | 1000 | D | 2 | 0.0020 | 5 | 3 |
|  |  | N | 2 | 0.0020 | 5 | 15 |
| Model D3 | 200 | D | 1 | 0.0050 | 10 | 5 |
|  |  | N | 1 | 0.0050 | 10 | 30 |
| SUMO | 100 | D | 2 | 0.0200 | 50 | 25 |
|  |  | N | 1 | 0.0100 | 25 | 80 |
| Multicopter | 10 | D | 1 | 0.1000 | 250 | 125 |
|  |  | N | 1 | 0.1000 | 250 | 800 |
| Thermal cameras | 1 | D | 0.5 | 0.5000 | 1250 | 600 |
|  |  | N | 0.2 | 0.2000 | 500 | 1600 |

When the attention is turned to the smallest scales as the ones provided by the multicopter and the thermal cameras, of the order of 1 *to 10* m, we see that the standard deviation of the surface temperature is of the same order as at larger scales, providing very high estimations of the advection term. In fact, Mahrt (2000) indicates that these heterogeneities may be restricted to the roughness sublayer ("the layer below the surface layer") that extends to the blending height, above which the effect of these small-scale heterogeneities is perceived as integrated by the surface layer. In the roughness layer, typically of the order of magnitude of the roughness elements of the surface, Monin-Obukhov similarity is not applicable because the turbulence is not in equilibrium with the local gradient. These roughness elements are of the order of a few centimetres in most of the small square.

This allows us to exclude the heterogeneity of very small scales from our analysis as detected by the multicopter and the thermal cameras that, in fact, overpass by far the values of the imbalances at day and at night. In practical terms, it also means that these thermal differences in the roughness layer do not manage to organize persistent circulations at the level of the measuring screen. Instead, the persistence of such heterogeneities may indicate that circulations between the centimetre and the metre scales very close to the ground may establish, which could contribute to the fact that the surface does not experience nocturnal runaway cooling, contrarily to what models generate in flat areas that they treat as homogeneous. This subject should be explored further in an independent research action.

Therefore, the most relevant range of scales are those comprising the hectometre and the decametre scales. The former ones correspond to the actual scales of landscape heterogeneities in the area, such as crop fields and wooden areas in between, or farms and small villages. Even the town of Lannemezan is structured in areas of characteristic sizes smaller than one kilometre. These patterns are either permanent (wooden areas, farms and villages) or slowly varying with the seasons (crops and grass lands). That is, these heterogeneities are fixed at the daily scale, and generate circulations that may persist for several hours and cannot be treated as turbulence. The estimations provided by the model and the SUMO indicate that these circulations may easily account for advections of the order of 10 W m$^{-2}$ that explain less than 10% of the imbalance in the daytime, but may be or the order of 30% in the nighttime, and as large as the other main terms in equations (1) and (2).

The scales of the order of a decametre, illustrated here with the multicopter data, indicate that the heterogeneities are large in the daytime, very much in accordance with the picture provided by LES and DNS of the Convective Boundary Layer (Van Heerwaarden et al. 2014), where small plumes exist everywhere in the first 10 m above the ground and only a few plumes (at a scale close to 100 m) manage to grow and make part of the mixed layer. It is difficult then to consider conceptually this variability as a contribution to the advection term, although there is no reason not to be able to compute the advection term, and in fact it may be behind some of the imbalance, explaining some tens of W m$^{-2}$ in the daytime.

The heterogeneities in the surface temperature at the decametre scale in the nighttime as seen by the multicopter are weak, of the order of 0.2 K, a value that we considered not relevant when found for the air temperature at 65 m as sampled by the SUMO. Therefore, even if the estimated advection term could explain largely the imbalance, we prefer to refrain from making any strong statement about this issue due to the few data available at night and conclude that more measurements are needed, also indicating that these scales may generate motions that could be included in the turbulence fluxes.

As the main point, it seems relevant to state that for scales of the order of hectometres, the circulations generated by surface heterogeneities may be relatively persistent and explain a substantial part of the imbalance in the SEB, especially at night.

For larger scales the contributions are small, whereas for finer scales the subject is still open to discussion, but probably these motions are small and restricted to very close to the surface or taken into account in the turbulence. Since the surface temperature field seems to have a variability close to that of air temperature at the screen level, a possible estimation of the contribution of the subgrid or subpixel variability to the SEB might be provided using $\sigma(LST)$ from satellite images in the approximate equation (4).

## 6 Conclusions

This work has explored the order of magnitude of the advection term in the SEB using broad estimations of the surface-layer thermal variability provided by a number of sources, including model outputs at different resolutions, satellite images, remotely-controlled measuring devices (SUMO and multicopter) and thermal cameras. The SEB is computed using the measurements on a small squared area in BLLAST, which provides an estimation of the imbalances, which is of the order of 200 W m$^{-2}$ in the central part of the day, and close to 30 W m$^{-2}$ at night, both values being very similar to the turbulent sensible and latent heat fluxes and the ground flux.

The variability of the surface temperature fields as provided by the different sources has been explored and it has been compared with the variability of the air temperatures in the surface when possible. It is seen that this variability has similar values for all the scales inspected, implying that the advection term is increasingly larger as the scale becomes finer. The variability of the air temperature close to the surface is similar to that at the surface, using the information that we have, essentially from the model outputs and the multicopter transects.

The advection term corresponding to scales greater than a kilometre are much smaller than the other terms and cannot explain any significant part of the imbalance, either because there are no real circulations performing the transport or because the steady state regime makes the net advection very small. On the other extreme of the spectrum of scales, those of the order of a metre still show very significant temperature variability, but the associated values of advection are too high to be meaningful, and probably are related to redistribution of heat in the first centimetres above the surface within the conceptual box of computation of the SEB and therefore not relevant for the SEB.

The current analysis points to the hypothesis that long-lasting terrain heterogeneities at the hectometre scale, like cultivated fields or small woods typical for the area, may generate motions that last longer than the averaging time of the turbulent fluxes and explain a significant part of the imbalance. Instead, the contribution of motions generated at the decametre or the metre scale, usually within the Surface Layer, provide unrealistically high values of the imbalance indicating that most likely they are already taken into account in the turbulent fluxes. To proceed towards more conclusive evidence of these qualitative results, specifically designed experiments should be conducted, providing better quantitive estimations and informing about the sign of the advection term.

*Acknowledgements.* BLLAST field experiment was made possible thanks to the contribution of several institutions and supports: INSU-CNRS (Institut National des Sciences de l'Univers, Centre National de la Recherche Scientifique, LEFE-IDAO program), Météo-France, Observatoire Midi-Pyrénées (University de Toulouse), EUFAR (EUropean Facility for Airborne Research) and COST ES0802 (European Cooperation in the field of Scientific and Technical). The field experiment would not have occurred without the contribution of all participating European and American research groups, which have all contributed in a significant amount (see supports). BLLAST field experiment was hosted by the instrumented site of Centre de Recherches Atmosphériques, Lannemezan, France (Observatoire Midi-Pyrénées, Laboratoire d'Aérologie). BLLAST data are managed by SEDOO, from Observatoire Midi-Pyrénées. We wish to particularly acknowledge Felipe Molinos, who assisted the University of the Balearic Islands team in the field, and the students of the Wageningen University that took the pictures with the thermal camera, specially Linda Kooijmans and Daniel Kunne. ECMWF and AEMET have provided computing time through the research project "Effect of the surface heterogeneities in the atmospheric boundary layer". The Spanish Ministry of Research has partially funded this action through grants of the Spanish Government CGL2009-12797-C03-01, CGL2012-37416-C04-01, supplemented with FEDER funds, CGL2015-65627-C3-1-R and PCIN-2014-016-C07-01, the latter part of the NEWA ERA-Net+ project of the European Union.

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
