# Peer review of "Estimation of the advection effects induced by surface heterogeneities in the surface energy budget"

_Atmospheric Chemistry and Physics, 2015_

## Referee Comment (RC1) · V. Caselles (Referee) · 3 Feb 2016

In my opinion the paper can be accepted with the following minor comment:

In Table 1, the authors must indicate the total error not the standard deviation. The total error, e, is

$$e = e_i + s$$

where $e_i$ is the sensitivity error (minimum amount which can be measured by the instrument used in each case), and s is the standard deviation.

---

## Referee Comment (RC2) · Anonymous Referee #2 · 1 Mar 2016

The investigation of the energy budget closure of the micrometeorological measurements and quantifying the effects of scale dependent advection are important from theoretical and practical points of view. The well written manuscript after the minor correction is suitable for publication.

Please also note the supplement to this comment:
http://www.atmos-chem-phys-discuss.net/acp-2015-1051/acp-2015-1051-RC2-supplement.pdf

**Supplement:**

**J. Cuxart et al., 2015: Estimation of the advection effects induced by surface heterogeneities in the surface energy budget**

1. *Does the paper address relevant scientific questions within the scope of ACP?*
   Yes. Investigation of surface energy budget closure and estimation of the sources of imbalance, first of all the scale dependent advection, are up-to-date questions.
2. *Does the paper present novel concepts, ideas, tools, or data?*
   Yes. Quantification of scale dependent advection term for SEB.
3. *Are substantial conclusions reached?*
   Partly. The conclusions are important from a methodological point of view. but there are not sufficiently many concrete results. I suggest giving more quantitative results.
4. Are *the scientific methods and assumptions valid and clearly outlined?*
   Yes. The principle of estimation of the scale-dependent advection term with the standard deviation of temperature fields is new but requires more detail explanations. How do you estimate the standard deviation of temperature in case of no advection? How does the advection depend on the standard deviation of temperature? I think it is not really a linear assumption. Is there any intercept?
5. *Are the results sufficient to support the interpretations and conclusions?*
   Yes
6. *Is the description of experiments and calculations sufficiently complete and precise to allow their reproduction by fellow scientists (traceability of results)?*
   Yes (Minor comments and question are given below.)
7. *Do the authors give proper credit to related work and clearly indicate their own new/original contribution?*
   Yes. The complexity of BLLAST campaign and the hierarchy of applied models give good material for the investigation of scale dependent advection.
8. *Does the title clearly reflect the contents of the paper?*
   Yes.
9. *Does the abstract provide a concise and complete summary?*
   Yes. (See the comments below.)
10. *Is the overall presentation well structured and clear?*
    Yes.
11. *Is the language fluent and precise?*
    Yes.
12. *Are mathematical formulae, symbols,* abbreviations, and units correctly defined and used?
    Yes*.*
13. *Should any parts of the paper (text, formulae, figures, tables) be clarified, reduced, combined, or eliminated?*
    Yes. A few comments are given below.
14. *Are the number and quality of references appropriate?*
    The most important references are in the manuscript. I suggest a few new references (see in the comments).
15. *Is the amount and quality of supplementary material appropriate?*
    Yes. The material of the paper is quite enough for understanding the messages of the authors.

**After minor corrections I suggest the publication of the manuscript.**

*Comments*

The investigation of the energy budget closure of the micrometeorological measurements and quantifying the effects of scale dependent advection are important from theoretical and practical points of view. The well written manuscript after the minor correction is suitable for publication.

*Abstract.*

Please give information about the eddy flux calculation methodology, uncertainty (for example in %) of calculation of energy budget components. Please give concrete results (numerical values) connected with the scale dependent effects of advection included surface heterogeneities.

Line 25-30
*Conceptually it is computed for a layer of infinitesimal depth across the interface in a horizontally homogeneous area, therefore* no storage or source terms are considered and, formally, the budget is expressed ...

Please clarify the sentence. I think the so called storage terms exist above horizontal homogeneous surfaces but negligabe in many cases as you mention later.

Please give the order of magnitude for additional terms in the energy budget equations above the short and tall vegetations (see for example photosynthesys, storage, etc.) Please give the estimation of order of these terms above the short and tall vegetations. (See for example Moderow et al., 2009 Theor Appl. Climatol 98. 397-412. for tall vegetation.)

Line 40-45
*These terrain heterogeneities may induce turbulent eddies and change the values of the turbulent heat flux compared to a completely homogeneous area.*

If it is possible please give a sentence about the underestimation of the available energy (H + LE) in the practice. The heterogeneity gives the reason of the changing the turbulent heat flux, but not enough explanation for the frequent underestimation of the fluxes.

Line 45-50
*... the advection terms can be computed using the divergence of temperature across the volume limits and the missing terms can be accounted for explicitly if the information is available (see Figure 1 in that paper).*

How do you interpret the effect of thermals and coherent structures in the inbalance (underestimate the fluxes in daytime)?

Line 57
Please give a few relevant citations.

Line 55-60
*... not considering the internal variability of the volume (in the air and in the soil) ...*

Please give more concrete information. What is meant by the variabilty of the volume air? Is it the form of the profile for example?

Line 55-60
*… such as water pumped up from below by the plant roots, ..*

Please clarify the effect and give a few citations.

Line 60-65

$$(Imb = TT + A + S + B + Ot).$$

Please use the same order as in equation (2).

Line 85-90, 135-140
*Hartogensis, (2015)*
*Wrenger et al, 2013*

Please give also the peer reviewed citations from the last few years.

Line 160-165
*Lafore et al, 1998*

Please check the year (1997 or 1998)

Line 160-165
*The run was from June 29th at 0000 UTC to July 3rd at 0000 UTC,*

Why didn't you used a spin up period for the mesoscale NH model? Please give more information about the data assimilation methodology: do you use any direct measurements in addition to the ECMWF model output?

Please give information about the differences of the turbulence parameterizations on the grid resolutions of 2 km, 400 m and 80 m. How many grid points were used in Domain 3?

Line 175-180

(Salomonson et al., 1989), (Martínez et al, 2010)

Please check the manuscript very carefully: et al or et al.?

Fig 1.
If it is possible please give a bigger map, for illustration of situation of D1 and please give also a more detailed map for D3 and surrounding. Please combine the 3 maps in one figure.

*Areas for which the average LST and its standard deviation given in Figure 3 are computed for model domains D1 and D2.*

Please clarify the sentences. What is meant by 'are those in green?

Line 190-195

$$\rho C p \Delta z u \approx 2500 \ J(K\,ms)^{-1},$$

Please give the uncertainty of the estimation (in %).

Line 200-205
*(Pietersen and De Coster, 2011)*

Please check the citation. I can see only De Coster and Pietersen, 2011.

Please give more detailed information about the flux calculation methodology (of the application spectral corrections, instrument specific corrections etc.) and the quality control of the fluxes.

Line 200-205
Why didn't you calculate directly the heat flux into the soil for each 30 min periods knowing the soil temperature profile, soil moisture and the soil heat flux at a depth of 5 cm?

Fig 3
*Top left: evolution of the average air temperature for some levels of the model ...*

Please clarify the sentence. For which model domain?

Please give the date in the bottom figures same as on Fig 1.

Line 240-245
Please give an explanation to the high standard deviations of $T_{surf\_MSG}$ 29 and 30 of June (upper right panel on Fig 3). Please give information about the comparison of model runs in D1 and D2 domains. What were the average temperature differences on the D2 model run using the 2 km and 400 m space resolution? If it is informative, please give a new figure.

Line 245-250
*Therefore, for scales larger than 1 km the expected contribution of the advection term to the SEB would be of the order of 10 $W\,m^{-2}$ in the daytime and of 5 $W\,m^{-2}$ at night,*

Please give information about the sign of the estimated advection in daytime and nighttime.

Fig 4.
Please give also the measurement interval.  How did you calculate the temperature field for the given time?

*The red rectangle indicates the position of the small square.*

What is meant by the small square? Please clarify the sentence.

If it is possible please give the scale in km in the figure of 1 and 4. This makes the analysis of the information easier.

Line 270-275
How do you estimate the temperature differences depending from the $\Delta x$ distance? How do you estimate the mean horizontal temperature gradient in equation 4? Please give more

detailed information about the methodology of the advection calculation based on the SUMO measurements.

*... with comparable values leading to similar estimations of A ...*

No point.

Fig 5.

`Difference Tsup between the square and outside`

What is *Tsup*? Please clarify the headline.

Please give the algorithm in more detail for the calculation of temperature differences in fig. 5. Do you use any weighing factor depending on the distance from the small square?

If it is possible please give information about the wind speed at 65m during the SUMO flights.

Line 285-290
*to 0.6 K from*

One space.

Line 290-295
Figure 7

Line 300-305
How do you estimate the sign of advection?

Line 305-310
Please give the type of the soil. Please clarify the soil moisture contents in%? What are the typical maximum and minimum soil moisture contents in this case?

Line 325-330
*The air temperature is sampled at 1 Hz, equivalent to a spatial resolution of a few meters.*

Please give information about the estimation of hysteresis of the measurements and the methodology of corrections.

Line 335-340
T(5 m)

One space.

Fig. 9.

Please give the definition of `Tsup 10 m`

I cannot see the abbreviation 'sup' in the text. Please give the explanation of the different colours in the top right figure. Please give the date and starting time for example.

*Nocturnal flight pattern and LST values (bottom left) and air temperature at 5 m a.g.l. (bottom right)*

Please give the date.

Line 340-345
*If we just take 0.5 K for the day and 0.2 K for the night, the corresponding advection values would be 100 and 40Wm$^{-2}$.*

Line 345-450
*up to 2 K variations*

Line 350-355
*being a factor that may oppose to runaway cooling as it is experienced in some numerical models ...*

If it is possible please give more concrete results about the measurements and the small scale modelling.

How do you estimate the sign of the advection?

Line 355-360
*Garai and Kleissi (2013)*

Please check the name Kleissi or Kleissl.

Line 361
Please give information about the soil (wet or dry). How do the measured inhomogeneities depend on the state of the soil? If it is possible please give a sentence?

Line 360-365
*We estimate the gradient of temperature $\Delta T/\Delta x$ as $\sigma(T)/r$, where r stands for the resolution.*

It is the key sentence. Please give more detailed explanation. How do you estimate the natural standard deviation of temperature? If the advection is negligible, $\sigma(T)$ goes to zero, is it true?

Table I.
Please give the height of temperature in term $\sigma(T)(K)$. (I think it is the surface.)

Line 375-380
*This is in agreement with the previous argumentations of Foken (2008) ...*

Please clarify the citations Foken 2008a or 2008b or both.

Line 65-70
*In this work we concentrate on the importance of the advection term A in the SEB which represents the effect of the motions of timescales longer that the turbulence-averaged ones.*

Please clarify the sentence and the mean goals of the paper because based on the discussion (see line 395, *Therefore the most relevant range of scales is the one comprising the hectometer and the decameter scales.*) the most relevant scales are 10-100 m, which are smaller than the calculated scale from the 30 min time scale with 1 m s$^{-1}$ characteristic wind speed.

Line 400-405
*… less than 10% …*

No space 10%.

Line 400-405
*… very much in accordance with the picture provided by LES and DNS of the Convective Boundary Layer, …*

Please give citations.

**References**

*Lafore, J. P., Stein, J., Asencio, N., Bougeault, P., Ducrocq, V., Duron, J., … Pinty, J. P. (1997). The Meso-NH atmospheric simulation system. Part I: adiabatic formulation and control simulations. In Annales Geophysicae (Vol. 16, No. 1, pp. 90-109). Springer-Verlag.*

*Leuning, R., Van Gorsel, E., Massman, W. J., Isaac, P. R. (2012). Reflections on the surface energy imbalance problem. Agricultural and Forest Meteorology, 156, 65-74. 480*

*Lothon, M., Lohou, F., Pino, D., Couvreux, F., Pardyjak, E. R., Reuder, J., … Augustin, P. (2014). The BLLAST field experiment: Boundary-layer late afternoon and sunset turbulence. Atmospheric chemistry and physics, 14(20), 10931-10960.*

*Oncley, S. P., Foken, T., Vogt, R., Kohsiek, W., DeBruin, H. A. R., Bernhofer, C., … Lehner, I. (2007). The energy balance experiment EBEX-2000. Part I: overview and energy balance. Boundary-Layer Meteorology, 123(1), 1-28.*

Please give the names of all authors.

Line 455-460

A turbulence scheme allowing for mesoscale and large?eddy simulations.

Please check the text.

Line 500-505

Bull. Amer. Meteor. Soc.

Please use the same format for the title of the journal. In other cases you used the full name, and no abbreviation.

---

## Author Comment (AC1) · 22 Apr 2016

Palma (Mallorca), 21st April 2016

Dear reviewers of paper acp-2015-1051

Thank you very much for reading our manuscript and for the improvements that you propose. Find below our answers point by point and the actions that we propose to do in the modified version of the manuscript. Proposed modified figures are also included in the supplement.

Yours sincerely, on behalf of all co-authors,

Joan Cuxart, corresponding author

**Reviewer 1**

***Request:*** Reviewer 1 asks that Table 1 is modified giving the total error, which is the sum of the standard deviation and the sensitivity error of each of the methods.

***Comment:*** This request raises interesting considerations. If we assumed, for the discussion, a typical instrumental error of 1 K to be added to sigma, the value of the estimation of the advection term would increase for all the considered sources due to the instrumental indetermination. However, this contribution is already taken into account in the SEB equation (2) in the term Ot, so there is the risk of double-counting this source in two terms (Adv and Ot). So we must, take both issues into account.

***Actions on the manuscript:*** To satisfy the well-justified reviewer request, the corresponding uncertainties of each method will be added in table I, the double counting issue will be mentioned, and the percent of the imbalance will be slightly increased, all of it not changing substantially the conclusions reached.

The suggested changes in the manuscript are, all in Section 5 (Discussion):
1) Add a paragraph, just after the first two in the section, that would read:

"An important issue to mention is that the uncertainties inherent to each method should be added to the value of the standard deviation. They are already conceptually taken into account in the term "Ot" of equation 2, but it is necessary to include this contribution to the variability of the measure in our estimations. The model, as seen in Figure 3, has an error for our case not larger than 1 K, as it is also the case for most remote sensing determinations of the surface temperature (see, e.g., Coll et al (1995) for MODIS). Thermal cameras report uncertainties of the order of 0.1 K. This fact is taken into account in Table 1."

2) Modify Table 1 as follows, for columns 1,4,5,6,7:
model D1 and MSG: sigma(T)_day=3, O(sigma/r)=0.00075, O(Adv)=2, %(imb)=1
model D1 and MSG : sigma(T)_night=2, O(sigma/r)=0.00005, O(Adv)=1, %(imb)=4
model D2 and MODIS; sigma(T)_day=2, O(sigma/r)=0.0002, O(Adv)=5, %(imb)=3
model D2 and MODIS; sigma(T)_night=2, O(sigma/r)=0.0002, O(Adv)=5, %(imb)=10
model D3, sigma(T)_day=1, O(sigma/r)=0.005, O(Adv)=10, %(imb)=5
model D3, sigma(T)_night=1, O(sigma/r)=0.005, O(Adv)=10, %(imb)=30
sumo, sigma(T)_day=2, O(sigma/r)=0.02, O(Adv)=50, %(imb)=25
sumo, sigma(T)_night=1, O(sigma/r)=0.01, O(Adv)=25, %(imb)=80
multicopter, sigma(T)_day=1, O(sigma/r)=0.1, O(Adv)=250, %(imb)=125
multicopter, sigma(T)_night=1, O(sigma/r)=0.1, O(Adv)=250, %(imb)=800
thermal cameras, sigma(T)_day=0.5, O(sigma/r)=0.5, O(Adv)=1250, %(imb)=600
thermal cameras, sigma(T)_night=0.2, O(sigma/r)=0.2, O(Adv)=500, %(imb)=1600

***Reference:*** Coll, C., Caselles, V., Galve, J. M., Valor, E., Niclos, R., Sánchez, J. M., & Rivas, R. (2005). Ground measurements for the validation of land surface temperatures derived from AATSR and MODIS data. *Remote Sensing of Environment, 97*(3), 288-300.

***Impact on the manuscript***: None of the new values alters the comments and the conclusions stated, which indicate that the estimation is relatively small for scales above the kilometre, too high for scales around the decametre or lower and potentially significant for scales around the hectometre.

**Reviewer 2**

*A) Answers to comments in the opening part*

**Reviewer's point #2:** Are substantial conclusions reached? Partly. The conclusions are important from a methodological point of view. but there are not sufficiently many concrete results. I suggest giving more quantitative results.

**Answer:** The paper is intended to be a first step to the evaluation of the order of magnitude of the advection with the available data. Given the actual distribution of stations in BLLAST it would be perhaps too bold to go further. Currently a new experiment is running on Majorca to provide better estimations of this term using a display of ten stations in a 1 km-squared, to develop the methodology further.

**Action on the paper:** The argument above will be stressed in the Introduction and in the Discussion

**Reviewer's point #3:** Are the scientific methods and assumptions valid and clearly outlined? Yes. The principle of estimation of the scale-dependent advection term with the standard deviation of temperature fields is new but requires more detail explanations. How do you estimate the standard deviation of temperature in case of no advection? How does the advection depend on the standard deviation of temperature? I think it is not really a linear assumption. Is there any intercept?

**Answer:** The standard deviation is estimated independently of advection by inspecting the variability of the recorded temperature in the area of interest. The increment of temperature is substituted by sigma(T) based on the data provided by the SUMO UAV, that shows that both magnitudes behave comparably and have similar orders of magnitude.

**Action on the paper:** See answer to DC21 below.

*B) Answers to detailed comments*

**Reviewer detailed comment #1 (DC1):** Abstract :Please give information about the eddy flux calculation methodology, uncertainty (for example in %) of calculation of energy budget components. Please give concrete results (numerical values) connected with the scale dependent effects of advection included surface heterogeneities.

**Answer and action on the paper:** In the abstract we will add that *turbulent fluxes are computed using the eddy-correlation method*. The estimations of the uncertainties will be given in Section 3, where the SEB is discussed (see also answer to DC15 on how uncertainties of turbulent fluxes are introduced in that Section).

**DC2:** Line 25-30: Conceptually it is computed for a layer of infinitesimal depth across the interface in a horizontally homogeneous area, therefore no storage or source terms are considered and, formally, the budget is expressed ...

R: Please clarify the sentence. I think the so called storage terms exist above horizontal

homogeneous surfaces but negligable in many cases as you mention later. Please give the order of magnitude for additional terms in the energy budget equations above the short and tall vegetations (see for example photosynthesys, storage, etc.) Please give the estimation of order of these terms above the short and tall vegetations. (See for example Moderow et al., 2009 Theor Appl. Climatol 98. 397-412. for tall vegetation.)

*Answer:* most of these comments have been made in the JGR paper by Cuxart et al (2015) and a similar discussion is given in the recent book by Moene and Van Dam (2014).

*Action in the paper*: add a short line refering to the subject and leading the reader to that paper and the reference therein: *"Conceptually, as described in Moene and Van Dam (2014) or Cuxart et al (2015), it is computed ..."*

*Reference:* Moene, A. F., & van Dam, J. C. (2014). Transport in the atmosphere-vegetation-soil continuum. Cambridge University Press.

**DC3:** Line 40-45: These terrain heterogeneities may induce turbulent eddies and change the values of the turbulent heat flux compared to a completely homogeneous area.

R: If it is possible please give a sentence about the underestimation of the available energy (H + LE) in the practice. The heterogeneity gives the reason of the changing the turbulent heat flux, but not enough explanation for the frequent underestimation of the fluxes.

*Answer*: We will add a short line commenting the loss of flux by the sensors and refer again to Cuxart et al (2015).

*Action in the paper:* Add in line 45 *"Nonetheless, errors in the determination of the turbulent fluxes must be kept in mind, very often implying an understimation of their value, due to the non-capturing of certain scales by the measuring devices (Foken 2008a)."*

**DC4:** Line 45-50 ... the advection terms can be computed using the divergence of temperature across the volume limits and the missing terms can be accounted for explicitly if the information is available (see Figure 1 in that paper).

R: How do you interpret the effect of thermals and coherent structures in the inbalance (underestimate the fluxes in daytime)?

*Answer:* in fact, one of the conclusions of the paper is that coherent structures lasting longer than the averaging time may be partly behind the lack of closure through the advection term. We will stress more the idea in the Conclusions section.

*Action in the paper*: In line 50 add *"For instance coherent structures lasting longer than the averaging time used to determine the averages and the fluxes may be contributing to the advection term"*

**DC5:** Line 57R: Please give a few relevant citations.

*Answer and action in the paper:* As mentioned, we will include the reference of Moone and Van Dam and refer to the references therein.

**DC6:** Line 55-60 : … not considering the internal variability of the volume (in the air and in the soil) ...

R: Please give more concrete information. What is meant by the variabilty of the volume air? Is it the form of the profile for example?

*Answer and action in the paper:* it is meant the material variations of the media, for instance objects over land, or soil heterogeneity. We will change that expression to *"... internal variability inside the volume, such as presence of objects over the ground or soil heterogeneity."*

**DC7:** Line 55-60 : … such as water pumped up from below by the plant roots, ..

R: Please clarify the effect and give a few citations.

*Answer and action in the paper:* it means that roots may bring water, and therefore transpiration, from depths outside the volume of interest. The sentence will be modified as *"...such as water pumped from below the volume of interest by plant roots (Moene and Van Dam, 2014)..."*

**DC8:** Line 60-65 R: Please use the same order as in equation (2).

*Answer and action in the paper:* It will be changed.

**DC9:** Line 85-90, 135-140 : Hartogensis, (2015) Wrenger et al, 2013. R: Please give also the peer reviewed citations from the last few years.

*Answer:* Unfortunately, peer-reviewed papers have not been produced yet for these works, they are in process.

**DC10:** Line 160-165: Lafore et al, 1998. R: Please check the year (1997 or 1998).

*Answer and action in the paper:* It is 1998. It will be changed in the reference list.

**DC11:** Line 160-165: The run was from June 29th at 0000 UTC to July 3rd at 0000 UTC,

R: Why didn't you used a spin up period for the mesoscale NH model? Please give more information about the data assimilation methodology: do you use any direct measurements in addition to the ECMWF model output? Please give information about the differences of the turbulence parameterizations on the grid resolutions of 2 km, 400 m and 80 m. How many grid points were used in Domain 3?

*Answer:* The model is solely initialised with ECMWF analysis, and the first 6 hours are usually discarded for analysis, since they are considered to be in the spin-up phase of the simulation. All domains use the same turbulence parameterisation, that is a 1d-parameterization. This is legitimate for D2 and D3 because the runs are for the nighttime and the turbulence is of smaller size. than the grid mesh. D3 domain has 250 times 250 points.

*Actions in the paper:* I) add *"...to July 3rd at 0000 UTC, considering the first six hours as the spin up period"*; ii) in line 170 add " *The model uses a standard one-dimensional 1.5 order scheme in the three domains...*"; iii) in line 168 *" D3 (250 times 250 points)"*

**DC12:** R:Please check the manuscript very carefully: et al or et al.?

*Answer and action in the paper:* we will write "et al." everywhere.

**_DC13_** R: I) Fig 1.If it is possible please give a bigger map, for illustration of situation of D1 and please give also a more detailed map for D3 and surrounding. Please combine the 3 maps in one figure.

ii) "Areas for which the average LST and its standard deviation given in Figure 3 are computed for model domains D1 and D2." R: Please clarify the sentences. What is meant by 'are those in green?

**_Answers and actions in the paper:_** I) we will rename Figure 1 as Figure 1a and add the figure below as Figure 1b, with caption *"Extension and topography of domains D2 and D3 (inside the purple rectangle). The position of Lannemezan is indicated with a cross"* ii) Areas coloured in green are those between 50 and 700 as for which standard deviation of LST is computed, as explained at the beginning of Subsection 4.1. This information will now be added in the caption of Figure 1.

[Figure]

**_DC14:_** R: Please give the uncertainty of the estimation (in %) for (rho Cp Delta(z) approx 2500 J/K/m/s)

**_Answer:_** the uncertainty is very large, this is why we use the symbol "approx". Taking a fixed (arbitrary) Delta(z) it lies mostly on the value of the wind speed. For clear days with weak winds, these values are usually between 1 and 2 m/s at a height of two meters above the ground. Therefore the uncertainty would be of the order of 100% and this is the main reason to work with orders of magnitudes instead of approximate values, since uncertainties would become too large.

**_Action in the paper:_** add in line 194 "*… of the advection -with large uncertainties of the order of 100 % due to the broad assumptions made-:* "

**_DC15:_** I) Line 200-205 (Pietersen and De Coster, 2011) R: Please check the citation. I can see only De Coster and Pietersen, 2011.

**_Answer and action in the paper:_** You are right. It is De Coster/Pietersen. It will be changed

in line 201.

ii) R: Please give more detailed information about the flux calculation methodology (of the application spectral corrections, instrument specific corrections etc.) and the quality control of the fluxes.

*Answer:* We do use fluxes from the standardized flux data base of BLLAST as described in that reference. We already list the basic methods that they use through a list of relevant references by Wilczak et al. (planar fit) and Webb et al, for the density correction. They also proceed to check correctness of record timing, they de-spike and make quality-control of the data. They also provide estimations of the error of the fluxes, usually circa 10%.

Action in the paper: add in line 203 " *...(Webb et al, 1980). Errors in the values of the turbulent fluxes are estimated to be in the order of 10%. Furthemore, correctness of the record timing is checked, and de-spiking and quality control are made in the ensemble of the BLLAST data set"*

**DC16:** Line 200-205 R: Why didn't you calculate directly the heat flux into the soil for each 30 min periods knowing the soil temperature profile, soil moisture and the soil heat flux at a depth of 5 cm?

*Answer:* the temperature measurement in the upper part of the soil was providing strange results and we decided not to use it.

***Action in the paper:*** we will include in line 205, *"and, due to unrealistic recorded values of the upper soil temperature, corrected to the values at the surface using..."*

**DC17:**.Fig 3 Top left: evolution of the average air temperature for some levels of the model ...  R: Please clarify the sentence. For which model domain?

***Answer and action in the paper:*** it is mentioned in the caption that it is for D1. We will add "model domain" after "D1" to improve clarity.

**DC18:** Line 240-245 R: I) Please give an explanation to the high standard deviations of $T_{surf\_MSG}$ 29 and 30 of June (upper right panel on Fig 3). ii) Please give information about the comparison of model runs in D1 and D2 domains. What were the average temperature differences on the D2 model run using the 2 km and 400 m space resolution? If it is informative, please give a new figure.

*Answers:* I) these large values were due to the presence of cloudiness that part of the day; ii) a new line (green dots) has been included in figures 3 bottom corresponding to the average and the sigma values for domain D1. No significative differences are seen in the averages and sigma is larger at D1 than at D2

***Actions in the paper:*** I) add in line 239: *Large puntual values of sigma(T) are due to cloudiness in June  29*, ii) The figures 3 bottom will be substituted by the ones below and in line 246, *"No significant differences are observed between D1 and D2 averaged values, but the standard deviation is higher at lower resolution, indicating that finer resolved scale motions may contribute to relax surface temperature variability"*

[Figure]

**DC19:** Line 245-250: *"Therefore, for scales larger than 1 km the expected contribution of the advection term to the SEB would be of the order of 10 W m$^{-2}$ in the daytime and of 5 W m$^{-2}$ at night"*. R: Please give information about the sign of the estimated advection in daytime and nighttime.

**Answer:** the sign of the advection largely depends on the sign of the wind, which, to our effects, is arbitrary and we decide not to discuss about it. In 95% of the cases (see Cuxart et al, 2015), the imbalance misses energy, and we assume in this study that the advection effects will be a contribution trying to explain part of it. A short line is added in the manuscript about this issue.

**Action in the paper:** Add in line 248 *"The sign of the advection term would result of the inspection of the wind direction between heterogeneities. We do not have detailed information at this stage and we restrict ourselves to discuss the order of magnitude of the term, expecting that normally it will be a contribution tending to reduce the imbalance."*

**DC20:** Fig 4. R: i) Please give also the measurement interval. How did you calculate the temperature field for the given time?; ii) "The red rectangle indicates the position of the small square", what is meant by the small square? Please clarify the sentence; iii) If it is possible please give the scale in km in the figure of 1 and 4. This makes the analysis of the information easier.

**Answers and actions in the text:** i) SUMO typically sampled at 1 Hz, meaning an effective LST resolution near 100 m when combined with the field of view of the camera from a height of 70 m; ii) the "small square" as introduced in section 2, is the flat 160m *160 m area where the surface based measurements were made; iii) figure 1 is better in lat/lon because of Earth's spherical form; for figure 4 equivalence to kilometres will be provided in the caption.

**DC21:** Line 270-275 R: How do you estimate the temperature differences depending from Delta(x) distance? How do you estimate the mean horizontal temperature gradient in equation 4? Please give more detailed information about the methodology of the advection calculation

based on the SUMO measurements.

*Answers:* SUMO flies over a standard heterogeneous area in the Lannemezan Plateau, passing also over the "small square" where surface data are taken, which is singular compared to its surroundings, because it is squared, covered by a mixture of recently cut grass and some spots of bare soil, whereas the surroundings include small wooden areas and many grown agricultural fields. We firstly estimate Delta(LST) as the difference between average LST in the small red square and the average LST for the whole SUMO square in fig 4. Then we compute sigma(LST) from all the measurements over the SUMO square and we see that it compares very well with Delta(LST) in terms of time evolution and proportionality (see figs 5 left and 6 left). This allows us to make the strong hypothesis that we can substitute Delta(LST) by sigma(LST), sustained also by the fact that we are only interested in orders of magnitude, not in accurate estimations of its effect. Later we divide sigma(LST) by the resolution of the device to estimate delta(LST)/delta(x) by sigma(LST)/resolution. The final strong hypothesis is to assume that LST is a good surrogate of T(2m). Obviously all these hypotheses make the results only a first guess that must be confirmed by further studies, more precise, currently under way. In principle all this information is already given in the paper, but we will try to make the rationale more clear.

*Action in the paper:* substitute paragraph 273-277 by the above explanation, written more succintly.

*DC22:* Fig 5. i) R: What is Tsup? Please clarify the headline; ii) Please give the algorithm in more detail for the calculation of temperature differences in fig. 5. Do you use any weighing factor depending on the distance from the small square?; iii) If it is possible please give information about the wind speed at 65m during the SUMO flights.

*Answers:* i) Tsup is LST. It is described in the caption; ii) see previous answer (21); iii) these points comprise all the days when SUMO could fly, typically wind varied at that height between 2 and 5 m/s, but we do not see the point, more than a variation of LST resolution, which is already estimated broadly.

*DC23:* Line 300-305 R: How do you estimate the sign of advection?

*Answer:* see answer to DC19

*DC24:* Line 305-310 R: I) Please give the type of the soil. ii) Please clarify the soil moisture contents in%? What are the typical maximum and minimum soil moisture contents in this case?

*Answer:* the type of soil is mostly clay, sometimes bare, more often covered by a layer combining dead and alive vegetation. The units of soil moisture are percent of volume. Saturation contents is the one shown in figure 8 top left (just after intense rain). We ignore the minimum value, but the upper part dried very quickly and took very low values.

*Action in the paper:* I) In line 310 "*The soil is mostly clay, many times covered by a mixture of vegetation dead and alive*"; ii) In caption of Figure 8 add:"*The soil moisture is given in percent of volume*"

*DC25:* Line 325-330 *"The air temperature is sampled at 1 Hz, equivalent to a spatial resolution of a few meters."* R: Please give information about the estimation of hysteresis of

the measurements and the methodology of corrections.

*Answer:* Flights were made at very low speed and a delay correction was applied to compensate for the relatively slow response time of the sensor.

*Action in the paper:* Add in line 328 "The slow response time can be compensated by a numerical correction scheme which assumes a linear response of the sensor for the difference between instantaneous measured parameter (here: air temperature) and the true ambient value of this parameter (Reuder et al, 2009)

*Reference:* Reuder, J., Brisset, P., Jonassen, M., Müller, M., & Mayer, S. (2009). The Small Unmanned Meteorological Observer SUMO: A new tool for atmospheric boundary layer research. *Meteorologische Zeitschrift*, *18*(2), 141-147.

*DC26:* Fig. 9. R: Please give the definition of Tsup 10m. I cannot see the abbreviation 'sup' in the text. Please give the explanation of the different colours in the top right figure. Please give the date and starting time for example. Nocturnal flight pattern and LST values (bottom left) and air temperature at 5 m a.g.l. (bottom right). Please give the date.

*Answers and action in the paper*: i) Tsup 10 means LST as sampled from a height of 10 m agl; ii) the different colors in fig 9 (top right) correspond to 4 different profiles made nearby in the small square, all made within a couple of minutes. Both issues will be described in the figure caption.

*DC27:* Nocturnal flight pattern and LST values (bottom left) and air temperature at 5 m a.g.l. (bottom right). R: Please give the date.

*Answer and action in the paper*: the date is July 5th, 2011, 0325 UTC. We have realized that the figures 5 bottom left and bottom right were exchanged! We have now corrected this issue and given the data in the figure caption.

*DC28:* Line 340-345 *"If we just take 0.5 K for the day and 0.2 K for the night, the corresponding advection values would be 100 and 40Wm$^{-2}$."; Line 345-450 : "up to 2 K variations" ; Line 350-355 ; "being a factor that may oppose to runaway cooling as it is experienced in some numerical models ..."*

R: If it is possible please give more concrete results about the measurements and the small scale modelling. How do you estimate the sign of the advection?

*Answer:* These values for the multicopter are estimated from figure 9 and other similar figures not shown, and are only broad estimations. As stated before, a campaign is currently underway trying to provide better numerical estimations of this factors. Concerning the sign of advection, see again answer to DC19, but just let us mention that this particular issue will also be addressed in the new campaign.

*Action in the paper:* In line 342: *"If we estimate the values from Figure 9 and take Delta(x) as 0.5 K for the day and 0. 2 K for the night..."*

*DC29:* Line 355-360: Garai and Kleissi (2013) R: Please check the name Kleissi or Kleissl. *Answer:* KLEISSL

**_DC30:_**  Line 361 R: Please give information about the soil (wet or dry). How do the measured inhomogeneities depend on the state of the soil? If it is possible please give a sentence?

**_Answer_**: Soil was experiencing consecutive drying episodes, because there were rainy events about every 3 days. Therefore availability of soil moisture was high, even if the upper layer was drying progressively and relatively fast.

**_Action in the paper:_** In line 361 add  *"The moisture contents at the upper part of the soil may modulate these variations, but in general there was good availability of water in the upper part of the soil due to recent rain events"*.

**_DC31:_** Line 360-365 "We estimate the gradient of temperature $\Delta T/\Delta x$ as $\sigma(T)/r$, where r stands for the resolution. "

R: It is the key sentence. Please give more detailed explanation. How do you estimate the natural standard deviation of temperature? If the advection is negligible, $\sigma(T)$ goes to zero, is it true?

**_Answer:_** the basic explanation has been given in answer to DC19. Your last sentence is unclear to us. We are assuming that if there are local variations of temperature, and there is some wind moving them around, the corresponding thermal advection may bring or take away heat from the volume of interest. We would therefore say that, if wind is negligible or if the terrain is thermally homogeneous, then advection tends to zero, which seems to be very rarely the case.

**_Action in the paper:_** the one described in answer to DC19

**_DC32:_** Table I. R: Please give the height of temperature in term $\sigma(T)(K)$. (I think it is the surface.)

**_Answer and action in the paper:_** we describe in the text that it is hypothesized that T of air in the volume and LST have comparable variances. We will indicate this in the caption of the table.

**_DC33:_** Line 375-380: *"This is in agreement with the previous argumentations of Foken (2008) ..."*  R: Please clarify the citations Foken 2008a or 2008b or both.

**_Answer and action in the paper_**: Both. We will add both references in line 378.

**_DC34:_** Line 65-70 *"In this work we concentrate on the importance of the advection term A in the SEB which represents the effect of the motions of timescales longer that the turbulence-averaged ones. "*

R: Please clarify the sentence and the mean goals of the paper because based on the discussion (see line 395, Therefore the most relevant range of scales is the one comprising the hectometer and the decameter scales.) the most relevant scales are 10-100 m, which are smaller than the calculated scale from the 30 min time scale with 1 m/s characteristic wind speed.

**_Answer:_** You are right. We must stress that we refer to semi-permanent hectometer scales structures that last longer than 30', meaning those linked to well defined terrain

heterogeneities, such as adjacent fields with different thermal properties. It is already said (line 398 and line 416 and the following ones), but we will make it clearer.

***Action in the paper:*** revise wording of the last paragraphs of the Conclusions so that they read better.

**_DC35:_** Line 400-405 "… very much in accordance with the picture provided by LES and DNS of the Convective Boundary Layer, ... " R: Please give citations.

***Action in the paper***: In line 404 we cite the paper on DNS of the CBL by van Heerwaarden et al. (JAS, 2014):

***Reference:*** Van Heerwaarden, Chiel C., Juan Pedro Mellado, and Alberto De Lozar. "Scaling laws for the heterogeneously heated free convective boundary layer." Journal of the Atmospheric Sciences 71.11 (2014): 3975-4000.

**_DC36:_** Other minor issues (typos and similar): They will be all taken into account.

---

## Author Response (AR1)

Palma (Mallorca), 18th May 2016

Dear editor and reviewers of paper acp-2015-1051

This document updates the "answers to the reviewers" sent some weeks ago, detailing the final actual changes made in the revised paper. In the companion file, you may read the revised version of the manuscript with the changed parts highlighted (in bold italic).

Yours sincerely, on behalf of all co-authors,

Joan Cuxart, corresponding author

**Updated response to the reviewers, including the actual final changes in the revised version**

**Reviewer 1**

***Request:*** Reviewer 1 asks that Table 1 is modified giving the total error, which is the sum of the standard deviation and the sensitivity error of each of the methods.

***Comment:*** This request raises interesting considerations. If we assumed, for the discussion, a typical instrumental error of 1 K to be added to sigma, the value of the estimation of the advection term would increase for all the considered sources due to the instrumental indetermination. However, this contribution is already taken into account in the SEB equation (2) in the term Ot, so there is the risk of double-counting this source in two terms (Adv and Ot). So we must, take both issues into account.

***Actions on the manuscript:*** To satisfy the well-justified reviewer request, the corresponding uncertainties of each method will be added in table I, the double counting issue will be mentioned, and the percent of the imbalance will be slightly increased, all of it not changing substantially the conclusions reached.

The changes made in the manuscript are, all in Section 5 (Discussion):
1) A new paragraph, just after the first two in the section, that reads:
***"An important issue to mention is that the uncertainties inherent to each method should be added to the value of the standard deviation. They are already conceptually taken into account in the term Ot of equation 2, but it is necessary to include this contribution to the variability of the measure in our estimations. The model, as seen in Figure 3, has an error for our case not larger than 1 K, as it is also the case for most remote sensing determinations of the surface temperature (see, e.g., Coll et al (1995) for MODIS). Thermal cameras report uncertainties of the order of 0.1 K. This fact is taken into account in Table 1."***

2) The new table 1, which increases the value of the uncertainties by estimating the systematic error:

**Table 1.** Estimation of the order of magnitude of the advection term $A$ in the Surface Energy Budget equation, for different sources and scales, taking 200 W m$^{-2}$ as imbalance at the center of the day (D) and 30 W m$^{-2}$ at night (N), *also considering the error of each source*. The orders of magnitude are rounded, as are the percents of the imbalance. ***Standard deviation of LST values are used as surrogates of horizontal gradients of the Surface-Layer air temperature.***

| Source | Scale r (m) | D/N | $\sigma(T)(K)$ | $O(\sigma(T)/r)(K/m)$ | $O(Adv(T))(W\,m^{-2})$ | % Imb |
|---|---|---|---|---|---|---|
| Model D1 and MSG | 4000 | D | 3 | 0.00075 | 2 | 1 |
| | | N | 2 | 0.0005 | 1 | 4 |
| Model D2 and MODIS | 1000 | D | 2 | 0.0020 | 5 | 3 |
| | | N | 2 | 0.0020 | 5 | 15 |
| Model D3 | 200 | D | 1 | 0.0050 | 10 | 5 |
| | | N | 1 | 0.0050 | 10 | 30 |
| SUMO | 100 | D | 2 | 0.0200 | 50 | 25 |
| | | N | 1 | 0.0100 | 25 | 80 |
| Multicopter | 10 | D | 1 | 0.1000 | 250 | 125 |
| | | N | 1 | 0.1000 | 250 | 800 |
| Thermal cameras | 1 | D | 0.5 | 0.5000 | 1250 | 600 |
| | | N | 0.2 | 0.2000 | 500 | 1600 |

3) Add new reference: Coll, C., Caselles, V., Galve, J. M., Valor, E., Niclos, R., Sánchez, J. M., & Rivas, R. (2005). Ground measurements for the validation of land surface temperatures derived from AATSR and MODIS data. *Remote Sensing of Environment*, *97*(3), 288-300.

***Impact on the manuscript***: None of the new values alters the comments and the conclusions stated, which indicate that the estimation is relatively small for scales above the kilometre, too high for scales around the decametre or lower and potentially significant for scales around the hectometre.

**Reviewer 2**

*A) Answers to comments in the opening part*

**Reviewer's point #2:** Are substantial conclusions reached? Partly. The conclusions are important from a methodological point of view. but there are not sufficiently many concrete results. I suggest giving more quantitative results.

*Answer:* The paper is intended to be a first step to the evaluation of the order of magnitude of the advection with the available data. Given the actual distribution of stations in BLLAST it would be perhaps too bold to go further. Currently a new experiment is running on Majorca to provide better estimations of this term using a display of ten stations in a 1 km-squared, to develop the methodology further.

*Actions in the paper:*
a) In the Introduction: ***"At this point it is necessary to make clear that reliable quantitative conclusions are very difficult to obtain with the approach used in this work and the available data. However, comprehensive qualitative results will be obtained based on broad approximations and estimations of the order of magnitude of A depending on the scale analyzed. Therefore, we consider it a first methodological step opening the way to more precise and focused studies to come."***
b) In the Discussion: ***Let us recall that the main aim of this work, provided the available methods and data, is to provide comprehensive qualitative results for A for each analized scale, hoping that more precise experiments will be made in the near future.***
c) In the Conclusions: ***The current analysis points to the hypothesis that long-lasting terrain heterogeneities at the hectometre scale, like cultivated fields or small woods typical for the area, may generate motions that last longer than the averaging time of the turbulent fluxes and explain a significant part of the imbalance. Instead, the contribution of motions generated at the decametre or the metre scale, usually within the Surface Layer, provide unrealistic high values indicating that most likely they are already taken into account in the turbulent fluxes. To proceed towards more conclusive evidence of these qualitative results, specifically designed experiments should be conducted, providing better quantitive estimations and informing about the sign of the advection term.***

**Reviewer's point #3:** Are the scientific methods and assumptions valid and clearly outlined? Yes. The principle of estimation of the scale-dependent advection term with the standard deviation of temperature fields is new but requires more detail explanations. How do you estimate the standard deviation of temperature in case of no advection? How does the advection depend on the standard deviation of temperature? I think it is not really a linear assumption. Is there any intercept?

*Answer:* The standard deviation is estimated independently of advection by inspecting the variability of the recorded temperature in the area of interest. The increment of temperature is substituted by sigma(T) based on the data provided by the SUMO UAV, that shows that both magnitudes behave comparably and have similar orders of magnitude.

*Action in the paper:* The methodology of treatment of the advection term has been detailed step by step in section 3:
*For simplification purposes we will:*

*-neglect here the vertical advection (taking w = 0 in average is reasonable), implying that the error associated is included in the Ot term of the complete SEB;*

*-take 1 m s−1 as the order of magnitude of the wind in the Surface Layer in the clear skies and non windy cases subject of this study, regardless of its direction, therefore ignoring the sign of A;*

*-approximate the average horizontal surface temperature gradient in an area by the standard deviation of the surface temperature, supported by SUMO measurements, keeping in mind that we are concerned solely with orders of magnitude of A;*

*-consider the LST variability as a good estimation of the variability of the air temperature at the Surface Layer, as supported by the measurements of the multicopter;*

*-take the factor $\rho Cp\Delta zu \approx 2500 J(Kms)^{-1}$, where $\Delta z=2m$, leading to an expression for the order of magnitude of the advection term.*

*It is clear, from the large number of hypotheses made and its significance, that the results presented below will be broader estimations of the value of A for a given scale and source of information, with large uncertainties of the order of 100% or even above. However, these results will show significant differences in the orders of magnitude for the explored scales, allowing to reach some informative results. The approximate equation that we will use reads ...*

*B) Answers to detailed comments*

**Reviewer detailed comment #1 (DC1):** Abstract :Please give information about the eddy flux calculation methodology, uncertainty (for example in %) of calculation of energy budget components. Please give concrete results (numerical values) connected with the scale dependent effects of advection included surface heterogeneities.

*Answer and action in the paper:*
In the abstract: *"...the surface energy budget (SEB), for which the turbulent fluxes are computed using the eddy-correlation method. "*
In Section 3: *"Errors in the values of the turbulent fluxes are estimated to be in the order of 10%. Furthermore, correctness of the record timing is checked, and de-spiking and quality control are made in the ensemble of the BLLAST data set."*
Values related to the scale: provided in Table 1

***DC2:*** Line 25-30: Conceptually it is computed for a layer of infinitesimal depth across the interface in a horizontally homogeneous area, therefore no storage or source terms are considered and, formally, the budget is expressed ...

R: Please clarify the sentence. I think the so called storage terms exist above horizontal homogeneous surfaces but negligable in many cases as you mention later. Please give the order of magnitude for additional terms in the energy budget equations above the short and tall vegetations (see for example photosynthesys, storage, etc.) Please give the estimation of order of these terms above the short and tall vegetations. (See for example Moderow et al., 2009 Theor Appl. Climatol 98. 397-412. for tall vegetation.)

***Answer:*** most of these comments have been made in the JGR paper by Cuxart et al (2015) and a similar discussion is given in the recent book by Moene and Van Dam (2014).

***Action in the paper***: In the Introduction: ***"Conceptually, as described in Moene and Van Dam (2014) or Cuxart et al. (2015), it is computed for a layer of infinitesimal depth across the interface in a horizontally homogeneous area ..."***

***New reference:*** Moene, A. F., & van Dam, J. C. (2014). Transport in the atmosphere-vegetation-soil continuum. Cambridge University Press.

***DC3:*** Line 40-45: These terrain heterogeneities may induce turbulent eddies and change the values of the turbulent heat flux compared to a completely homogeneous area.

R: If it is possible please give a sentence about the underestimation of the available energy (H + LE) in the practice. The heterogeneity gives the reason of the changing the turbulent heat flux, but not enough explanation for the frequent underestimation of the fluxes.

***Action in the paper:*** In the Introduction: ***"Another important factor to consider is that instrumental errors in the determination of the turbulent fluxes must be kept in mind, very often implying an underestimation of their value, due to the non-capturing of certain scales by the measuring devices (Foken 2008a)."***

***DC4:*** Line 45-50 ... the advection terms can be computed using the divergence of temperature across the volume limits and the missing terms can be accounted for explicitly if the information is available (see Figure 1 in that paper).

R: How do you interpret the effect of thermals and coherent structures in the inbalance (underestimate the fluxes in daytime)?

***Answer:*** in fact, one of the conclusions of the paper is that coherent structures lasting longer than the averaging time may be partly behind the lack of closure through the advection term.

***Action in the paper***: In the Introduction: ***"Coherent structures lasting longer than this averaging time are most likely contributing significantly to this term, as would be the case for circulations between adjacent parcels of terrain at different temperatures, of a spatial scale still to be determined."***

In the Conclusions: ***"The current analysis points to the hypothesis that long-lasting terrain heterogeneities at the hectometre scale, like cultivated fields or small woods typical for the area, may generate motions that last longer than the averaging time of the turbulent fluxes and explain a significant part of the imbalance."***

***DC5:*** Line 57R: Please give a few relevant citations.

***Answer and action in the paper:*** As mentioned, the reference of Moone and Van Dam (2014) is included, and we refer to the references therein.

***DC6:*** Line 55-60 : … not considering the internal variability of the volume (in the air and in the soil) ...

R: Please give more concrete information. What is meant by the variabilty of the volume air? Is it the form of the profile for example?

***Answer and action in the paper:*** it is meant the material variations of the media, for instance objects over land, or soil heterogeneity. Sentence changed to ***"… such as not considering the internal variability of the volume, such as presence of objects over the ground or soil heterogeneity,"***

***DC7:*** Line 55-60 : … such as water pumped up from below by the plant roots, ..

R: Please clarify the effect and give a few citations.

***Answer and action in the paper:*** it means that roots may bring water, and therefore transpiration, from depths outside the volume of interest. The sentence has been modified as ***"...such as water pumped up from below the volume of interest by plant roots (Moene and Van Dam, 2014)..."***

***DC8:*** Line 60-65 R: Please use the same order as in equation (2).

***Answer and action in the paper:*** It has been changed.

***DC9:*** Line 85-90, 135-140 : Hartogensis, (2015) Wrenger et al, 2013. R: Please give also the peer reviewed citations from the last few years.

***Answer:*** Unfortunately, peer-reviewed papers have not been produced yet for these works, they are in process.

***DC10:*** Line 160-165: Lafore et al, 1998. R: Please check the year (1997 or 1998).

***Answer and action in the paper:*** After final checking we choose year 1997 (it was published end 1997 in a volume labelled 1998).

***DC11:*** Line 160-165: The run was from June 29th at 0000 UTC to July 3rd at 0000 UTC,

R: Why didn't you used a spin up period for the mesoscale NH model? Please give more information about the data assimilation methodology: do you use any direct measurements in addition to the ECMWF model output? Please give information about the differences of the turbulence parameterizations on the grid resolutions of 2 km, 400 m and 80 m. How many grid points were used in Domain 3?

***Answer:*** The model is solely initialised with ECMWF analysis, and the first 6 hours are usually discarded for analysis, since they are considered to be in the spin-up phase of the simulation. All domains use the same turbulence parameterisation, that is a 1d-parameterization. This is legitimate for D2 and D3 because the runs are for the nighttime and

the turbulence is of smaller size. than the grid mesh. D3 domain has 250 times 250 points.

***Actions in the paper:*** i) add ***"...to July 3rd at 0000 UTC, considering the first six hours as the spin up period"***; ii) in line 170 add ***" The model uses a standard one-dimensional 1.5 order scheme in the three domains..."***; iii) in line 168 ***" for a square of 250 grid points each side"***.

***DC12:*** R:Please check the manuscript very carefully: et al or et al.?

***Answer and action in the paper:*** we have now written "et al." everywhere.

***DC13*** R: I) Fig 1.If it is possible please give a bigger map, for illustration of situation of D1 and please give also a more detailed map for D3 and surrounding. Please combine the 3 maps in one figure.

ii) "Areas for which the average LST and its standard deviation given in Figure 3 are computed for model domains D1 and D2." R: Please clarify the sentences. What is meant by 'are those in green?

***Answers and actions in the paper:*** I) old Figure 1 is now Figure 1a and the figure below is new Figure 1b, with caption ***"Domains D2 and D3. The cross indicates the location of Lannemezan."*** ii) Areas coloured in green are those between 50 and 700 as for which standard deviation of LST is computed, as explained at the beginning of Subsection 4.1. In the caption it is now written: ***"Surface temperatures for areas with height above sea level between 50 and 700m (in green in Left Figure) are used to compute the average LST and its standard deviation."***

[Figure]

***DC14:*** R: Please give the uncertainty of the estimation (in %) for (rho Cp Delta(z) approx 2500 J/K/m/s)

***Answer:*** the uncertainty is very large, this is why we use the symbol "approx". Taking a fixed (arbitrary) Delta(z) it lies mostly on the value of the wind speed. For clear days with weak winds, these values are usually between 1 and 2 m/s at a height of two meters above the ground. Therefore the uncertainty would be of the order of 100% and this is the main reason

to work with orders of magnitudes instead of approximate values, since uncertainties would become too large.

*Action in the paper:* After detailing in section 3 the hypotheses made in the estimation of the advection term it is now stated: ***"It is clear, from the large number of hypotheses made and its significance, that the results presented below will be broader estimations of the value of A for a given scale and source of information, with large uncertainties of the order of 100% or even above. However, these results will show significant differences in the orders of magnitude for the explored scales, allowing to reach some informative results."***

***DC15:*** I) Line 200-205 (Pietersen and De Coster, 2011) R: Please check the citation. I can see only De Coster and Pietersen, 2011.

*Answer and action in the paper:* You are right. It is De Coster/Pietersen. It has been changed in line 201.

ii) R: Please give more detailed information about the flux calculation methodology (of the application spectral corrections, instrument specific corrections etc.) and the quality control of the fluxes.

*Answer:* We do use fluxes from the standardized flux data base of BLLAST as described in that reference. We already list the basic methods that they use through a list of relevant references by Wilczak et al. (planar fit) and Webb et al, for the density correction. They also proceed to check correctness of record timing, they de-spike and make quality-control of the data. They also provide estimations of the error of the fluxes, usually circa 10%.

*Action in the paper*: added in section 3: ***"Errors in the values of the turbulent fluxes are estimated to be in the order of 10%. Furthermore, correctness of the record timing is checked, and de-spiking and quality control are made in the ensemble of the BLLAST data set."***

***DC16:*** Line 200-205 R: Why didn't you calculate directly the heat flux into the soil for each 30 min periods knowing the soil temperature profile, soil moisture and the soil heat flux at a depth of 5 cm?

*Answer:* the temperature measurement in the upper part of the soil was providing strange results and we decided not to use it.

*Action in the paper:* we have included ***"and, due to unrealistic recorded values of the upper soil temperature, corrected to the values at the surface using..."***

***DC17:*** Fig 3 Top left: evolution of the average air temperature for some levels of the model ...  R: Please clarify the sentence. For which model domain?

*Answer and action in the paper:* it is mentioned in the caption that it is for D1. We have added "model domain" after "D1" to improve clarity.

***DC18:*** Line 240-245 R: I) Please give an explanation to the high standard deviations of $T_{surf\_MSG}$ 29 and 30 of June (upper right panel on Fig 3). ii) Please give information about the comparison of model runs in D1 and D2 domains. What were the average temperature differences on the D2 model run using the 2 km and 400 m space resolution? If it is

informative, please give a new figure.

***Answers:*** I) these large values were due to the presence of cloudiness that part of the day; ii) a new line (green dots) has been included in figures 3 bottom corresponding to the average and the sigma values for domain D1. No significative differences are seen in the averages and sigma is larger at D1 than at D2

***Actions in the paper:*** I) added : "*Note that large sporadic values of standard deviation for MSG on June 29 are due to cloud passages.*"; ii) The figures 3 bottom have been substituted by the ones below and the following text has been included: ***"No significant differences are observed between D1 and D2 averaged values, but the standard deviation is higher at lower resolution, indicating that finer resolved scale motions may contribute to relax surface temperature variability".***

[Figure]

***DC19:*** Line 245-250: "*Therefore, for scales larger than 1 km the expected contribution of the advection term to the SEB would be of the order of 10 W m$^{-2}$ in the daytime and of 5 W m$^{-2}$ at night*". R: Please give information about the sign of the estimated advection in daytime and nighttime.

***Answer:*** the sign of the advection largely depends on the sign of the wind, which, to our effects, is arbitrary and we decide not to discuss about it.

***Action in the paper:*** Added at the end of subsection 4.1: ***"The sign of the advection term would result of the inspection of the wind direction between heterogeneities. We do not have detailed information at this stage and we restrict ourselves to discuss the order of magnitude of the term."***

***DC20:*** Fig 4. R: i) Please give also the measurement interval. How did you calculate the temperature field for the given time?; ii) "The red rectangle indicates the position of the small square", what is meant by the small square? Please clarify the sentence; iii) If it is possible please give the scale in km in the figure of 1 and 4. This makes the analysis of the

information easier.

***Answers and actions in the text:*** i) SUMO typically sampled at ***1 Hz,*** meaning an effective LST resolution near 100 m when combined with the field of view of the camera from a height of 70 m; ii) the "small square" as introduced in section 2, is the flat 160m *160 m area where the surface based measurements were made; iii) figure 1 is better in lat/lon because of Earth's spherical form; for figure 4 the size of the red square is now indicated in the caption..

***DC21:*** Line 270-275 R: How do you estimate the temperature differences depending from Delta(x) distance? How do you estimate the mean horizontal temperature gradient in equation 4? Please give more detailed information about the methodology of the advection calculation based on the SUMO measurements.

***Answer and actions in the paper:*** the methodology has been clarified in section 3 and the two strong hypotheses are given here using SUMO data (using Delta (LST) as a surrogate of Delta(T)) and later in the multicopter part (using LST as a surrogate of air temperature in the surface layer).

In the SUMO part***: A very important result is that the standard deviation of LST ($\sigma$(LST ), Figure 6 left) for the complete SUMO square has a very similar time evolution as the one of the difference of temperatures between the small square and the average of the SUMO square (an estimation of $\Delta$(LST)). The factor of proportionality varies between 1 (in the morning and the evening) and 2 (at the centre of the day). Since we are concerned with orders of magnitude, a factor 2 allows to take $\sigma$(LST) as a surrogate of $\Delta$(LST). We shall keep this fact in mind, since we will apply it to some other sources based on this experimental evidence, recalling that the variability of LST is considered as an acceptable surrogate of the air temperature in the Surface Layer, as it will be seen later with the multicopter data.***

   In the multicopter part: ***The qualitative behaviour of the standard deviations of LST and the air temperature in the Surface Layer is very similar, allowing to take the variabilities of LST and air temperature at the Surface Layer as comparable when computing orders of magnitude, which is one of the major hypotheses of this work.***

***DC22:*** Fig 5. i) R: What is Tsup? Please clarify the headline; ii) Please give the algorithm in more detail for the calculation of temperature differences in fig. 5. Do you use any weighing factor depending on the distance from the small square?; iii) If it is possible please give information about the wind speed at 65m during the SUMO flights.

***Answers:*** i) Tsup is LST. It has been modified in the Figure.; ii) see answer (DC21); iii) these points comprise all the days when SUMO could fly, typically wind varied at that height between 2 and 5 m/s, but we do not see the point, more than a variation of LST resolution, which is already estimated broadly.

***DC23:*** Line 300-305 R: How do you estimate the sign of advection?

***Answer:*** see answer to DC19

***DC24:*** Line 305-310 R: I) Please give the type of the soil. ii) Please clarify the soil moisture contents in%? What are the typical maximum and minimum soil moisture contents in this case?

*Answer:* the type of soil is mostly clay, sometimes bare, more often covered by a layer combining dead and alive vegetation. The units of soil moisture are percent of volume. Saturation contents is the one shown in figure 8 top left (just after intense rain). We ignore the minimum value, but the upper part dried very quickly and took very low values.

*Action in the paper:* I) text added: **"The soil is mostly clay, with some bare spots, but mostly covered by grass (alive and dead)."**; ii) Added in caption of Figure 8 :**"The soil moisture is given in percent of volume".**

**DC25:** Line 325-330 *"The air temperature is sampled at 1 Hz, equivalent to a spatial resolution of a few meters."* R: Please give information about the estimation of hysteresis of the measurements and the methodology of corrections.

*Answer:* Flights were made at very low speed and a delay correction was applied to compensate for the relatively slow response time of the sensor.

*Action in the paper:* text added "**The slow response time can be compensated by a numerical correction scheme which assumes a linear response of the sensor for the difference between instantaneous measured parameter (here: air temperature) and the true ambient value of this parameter (Reuder et al, 2009)"**

*Reference:* Reuder, J., Brisset, P., Jonassen, M., Müller, M., & Mayer, S. (2009). The Small Unmanned Meteorological Observer SUMO: A new tool for atmospheric boundary layer research. *Meteorologische Zeitschrift*, *18*(2), 141-147.

**DC26:** Fig. 9. R: Please give the definition of Tsup 10m. I cannot see the abbreviation 'sup' in the text. Please give the explanation of the different colours in the top right figure. Please give the date and starting time for example. Nocturnal flight pattern and LST values (bottom left) and air temperature at 5 m a.g.l. (bottom right). Please give the date.

*Answers and action in the paper*: i) Tsup 10 in fact mean LST as sampled from a height of 5 m agl; ii) the different colors in fig 9 (top right) correspond to 4 different profiles made nearby in the small square, all made within a couple of minutes. Both issues are now described in the figure caption.

**DC27:** Nocturnal flight pattern and LST values (bottom left) and air temperature at 5 m a.g.l. (bottom right). R: Please give the date.

*Answer and action in the paper*: the date is July 5th, 2011, 0325 UTC. We have realized that the figures 5 bottom left and bottom right were exchanged! We have now corrected this issue and given the data in the figure caption.

**DC28:** Line 340-345 *"If we just take 0.5 K for the day and 0.2 K for the night, the corresponding advection values would be 100 and $40 Wm^{-2}$."; Line 345-450 : "up to 2 K variations" ; Line 350-355 ; "being a factor that may oppose to runaway cooling as it is experienced in some numerical models …"*

R: If it is possible please give more concrete results about the measurements and the small scale modelling. How do you estimate the sign of the advection?

*Answer:* These values for the multicopter are estimated from figure 9 and other similar

figures not shown, and are only broad estimations. As stated before, a campaign is currently underway trying to provide better numerical estimations of this factors. Concerning the sign of advection, see again answer to DC19, but just let us mention that this particular issue will also be addressed in a new campaign currently running at the Unicersity Campus at Mallorca.

*Action in the paper:* text added ***"Estimating the values from Figure 9 as Δx equal to 0.5 K for the day and 0. 2 K for the night..."***

**DC29:** Line 355-360: Garai and Kleissi (2013) R: Please check the name Kleissi or Kleissl. *Answer:* KLEISSL

**DC30:** Line 361 R: Please give information about the soil (wet or dry). How do the measured inhomogeneities depend on the state of the soil? If it is possible please give a sentence?

*Answer*: Soil was experiencing consecutive drying episodes, because there were rainy events about every 3 days. Therefore availability of soil moisture was high, even if the upper layer was drying progressively and relatively fast.

*Action in the paper:* text added ***"The moisture contents at the upper part of the soil may modulate these variations, but in general there was good availability of water in the upper part of the soil due to recent rain events"***.

**DC31:** Line 360-365 "We estimate the gradient of temperature ΔT/Δx as σ(T)/r, where r stands for the resolution. "

R: It is the key sentence. Please give more detailed explanation. How do you estimate the natural standard deviation of temperature? If the advection is negligible, σ(T) goes to zero, is it true?

*Answer:* the basic explanation has been given in answer to DC21, supplemented by extra info in answers to DC19 and main point number 3. Your last sentence is unclear to us. We are assuming that if there are local variations of temperature, and there is some wind moving them around, the corresponding thermal advection may bring or take away heat from the volume of interest. We would therefore say that, if wind is negligible or if the terrain is thermally homogeneous, then advection tends to zero, which seems to be very rarely the case.

*Action in the paper:* the one described in answer to DC21 and DC19.

**DC32:** Table I. R: Please give the height of temperature in term σ(T)(K). (I think it is the surface.)

*Answer and action in the paper:* we describe in the text that it is hypothesized that T of air in the volume and LST have comparable variances. We have indicated this in the caption of the table as: ***"Standard deviation of LST values are used as surrogates of horizontal gradients of the Surface-Layer air temperature"***.

**DC33:** Line 375-380: *"This is in agreement with the previous argumentations of Foken (2008) ..."* R: Please clarify the citations Foken 2008a or 2008b or both.

*Answer and action in the paper*: Both. We have added both references in this sintence.

***DC34:*** Line 65-70 *"In this work we concentrate on the importance of the advection term A in the SEB which represents the effect of the motions of timescales longer that the turbulence-averaged ones. "*

R: Please clarify the sentence and the mean goals of the paper because based on the discussion (see line 395, Therefore the most relevant range of scales is the one comprising the hectometer and the decameter scales.) the most relevant scales are 10-100 m, which are smaller than the calculated scale from the 30 min time scale with 1 m/s characteristic wind speed.

***Answer:*** You are right. It is now further stressed that we refer to semi-permanent hectometer scales structures that last longer than 30', meaning those linked to well defined terrain heterogeneities, such as adjacent fields with different thermal properties.

***Action in the paper:*** text added in the Introduction ***"Coherent structures lasting longer than this averaging time are most likely contributing significantly to this term, as would be the case for circulations between adjacent parcels of terrain at different temperatures, of a spatial scale still to be determined."***

and text added in the Conclusions: ***"The current analysis points to the hypothesis that long-lasting terrain heterogeneities at the hectometre scale, like cultivated fields or small woods typical for the area, may generate motions that last longer than the averaging time of the turbulent fluxes and explain a significant part of the imbalance. Instead, the contribution of motions generated at the decametre or the metre scale, usually within the Surface Layer, provide unrealistic high values indicating that most likely they are already taken into account in the turbulent fluxes. To proceed towards more conclusive evidence of these qualitative results, specifically designed experiments should be conducted, providing better quantitive estimations and informing about the sign of the advection term."***

***DC35:*** Line 400-405 "… very much in accordance with the picture provided by LES and DNS of the Convective Boundary Layer, ... " R: Please give citations.

***Action in the paper***: The reference is made now to the paper on DNS of the CBL by van Heerwaarden et al. (JAS, 2014):

***Reference:*** Van Heerwaarden, Chiel C., Juan Pedro Mellado, and Alberto De Lozar. "Scaling laws for the heterogeneously heated free convective boundary layer." Journal of the Atmospheric Sciences 71.11 (2014): 3975-4000.

***DC36:*** Other minor issues (typos and similar): They have all been taken into account.

---

## Author Response (AR2)

***Document containing comments on the corrections requested by Co-Editor Dr. Pardyjak on paper acp-2015-1051.***

First of all, in behalf of all co-authors, I would like to acknowledge the generous effort of Dr. Pardyjak in reading very carefully our manuscript and suggesting countless improvements on the English language, which have all been incorporated in the corrected version.

**Changes concerning the text:**

This version is not highlighting the differences in respect to the previous one because most of them were mainly on details. Let me comment here those that have some more scientific contents:

1) In page 4, comment 8 asked for a better writing of a sentence refering to the distribution of soil moisture in the small square. The new sentence, shorter, writes: "A gentle slope towards SW favors accumulation of water at this part of the small square after rainy events.".

2) In page 5, comment number 10, asks also for rewriting and we have adopted the Co-Editor's suggestion: " but it provides a reasonable starting point.".

3) In page 7, comment number 24 asks to explain what a "This" in the beginning of a sentence means. It is now converted to "This selection".

4) In page 7, comment number 44, asks for a clearer explanation on the different values of sigma(LST) between domains D1 and D2. The modified sentence reads: "While the values of the 1.5-m air temperatures are very close in both model domains, the standard deviation is larger in the domain at lower resolution. This is probably indicating that higher horizontal resolution is able to transport more efficiently heat differences originated at the surface level."

5) In page 8, comment number 3, there was a mistake, and temperatures were not a 2m but at 1.5m

6) In two places a reference for the "runaway cooling effect" is requested. Now the paper of Viterbo et al is cited in the first appearance and included in the reference list.

7) In page 14, comment number 10, a sentence on the effect of the Soil Moisture in the surface albedo has been added and a new reference listed: "Albedo may also change significantly with the changes of SM, decreasing as SM increases (Sugathan et al., 2014)."

8) In page 4 a new reference has been added concerning the multicopter, since there has been recently a paper published including some results using the same system (Jiménez et al, 2016)

9) In the discussion, the paragraph on uncertainties has been revised and made more compact. Now it reads: "An important issue to mention is that the uncertainties inherent to each method should be considered in Table 1, even if they are already conceptually taken into account in the term $Ot$ of equation 2. The model, as seen in Figure 3, has an error for our case not larger than 1 K, as it is also the case for most remote sensing determinations of the surface temperature (see, e.g., Coll et al. (1995) for MODIS). Thermal cameras report uncertainties of the order of 0.1 K. "

**Changes concerning the figures:**

a) In figures 5, 6, 7 and 9-up-left, titles have been removed and labels modified as requested.

b) Figures having a colorbar: the units of the colorbar have been included in the caption.

c) we have not included the points of the vertical multicopter profiles because they correspond to a different day of the figures shown in the lower panel, fearing that it may induce confusion. Instead we have indicated that the profiles correspond to a sunny afternoon in the figure caption.

d) We have capitalized the first letter of the month in Figure 8, and compacted the figure titles.

At Palma (Majorca), on July 14, 2016

J. Cuxart, corresponding author